# HoliTom 🐱 : Holistic Token Merging for Fast Video Large Language Models

**Kele Shao**[1,2,3], **Keda Tao**[1,3], **Can Qin**[4], **Haoxuan You**[5], **Yang Sui**[6], **Huan Wang**[3,*]

[1]Zhejiang University  [2]Shanghai Innovation Institute  [3]Westlake University
[4]Salesforce AI Research  [5]Columbia University  [6]Rice University
https://github.com/cokeshao/HoliTom

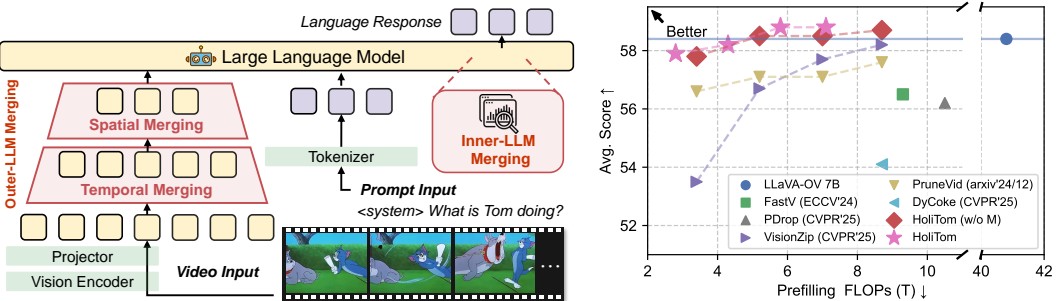

Figure 1: **Left:** We introduce *HoliTom*, a training-free holistic token merge method for fast video LLMs. Its key innovation lies in its global, redundancy-aware outer-LLM spatio-temporal compression and robust, token similarity-based inner-LLM compression. **Right:** The Efficiency/Performance trade-off curve of multiple training-free methods on four widely used video understanding benchmarks: MVBench, EgoSchema, LongVideoBench, and VideoMME. Our method, *HoliTom*, surpasses the SoTA approaches by maintaining 99.1% average performance while reducing FLOPs to 6.9%.

## Abstract

Video large language models (video LLMs) excel at video comprehension but face significant computational inefficiency due to redundant video tokens. Existing token pruning methods offer solutions. However, approaches operating within the LLM (inner-LLM pruning), such as FastV, incur intrinsic computational overhead in shallow layers. In contrast, methods performing token pruning before the LLM (outer-LLM pruning) primarily address spatial redundancy within individual frames or limited temporal windows, neglecting the crucial global temporal dynamics and correlations across longer video sequences. This leads to sub-optimal spatio-temporal reduction and does not leverage video compressibility fully. Crucially, the synergistic potential and mutual influence of integrating these strategies remain unexplored. To further reduce redundancy, we introduce *HoliTom*, a novel training-free holistic token merging framework. *HoliTom* employs outer-LLM pruning through global redundancy-aware temporal segmentation, followed by spatial-temporal merging to reduce visual tokens by over 90%, significantly alleviating the LLM's computational burden. Complementing this, we introduce a robust inner-LLM token similarity-based merging approach, designed for superior performance and compatibility with outer-LLM pruning. Evaluations demonstrate our method's promising efficiency-performance trade-off on LLaVA-OneVision-7B, reducing computational costs to 6.9% of FLOPs while maintaining 99.1% of the original performance. Furthermore, we achieve a 2.28× reduction in Time-To-First-Token (TTFT) and a 1.32× acceleration in decoding throughput, highlighting the practical benefits of our integrated pruning approach for efficient video LLMs inference.

---

*Corresponding authors: wanghuan@westlake.edu.cn

39th Conference on Neural Information Processing Systems (NeurIPS 2025).

# 1 Introduction

Video large language models (video LLMs) [25, 70, 56, 8, 9, 27, 30, 63, 67, 18, 51] show remarkable potential in understanding complex video content. However, their practical deployment is hindered by significant computational inefficiency. This inefficiency stems from processing high volumes of video tokens generated by encoding sampled frames, leading to substantial overhead, particularly due to the quadratic complexity of the attention mechanism in the LLMs. For videos with numerous frames, the input token count can easily reach tens of thousands, making inference computationally expensive. While prior works [6, 65, 61, 48, 19, 34, 36] have explored model compression and token pruning, achieving a desirable balance between efficiency and performance in video tasks remains an open challenge. Thus, developing effective methods to reduce video token redundancy while preserving critical semantic information is crucial for the widespread adoption of video LLMs.

Token pruning is a promising direction. These approaches generally fall into two categories depending on where pruning occurs. *Inner-LLM pruning* methods, such as FastV [6], TopV [64], and PDrop [61], operate within the LLM layers. However, they incur intrinsic computational and memory costs in the initial layers before pruning takes effect, limiting overall FLOPs reduction. *Outer-LLM pruning* methods process tokens before the main LLM computation. Some methods address spatial redundancy (VisionZip [65], PruMerge [42]), others tackle temporal aspects within limited temporal windows (DyCoke [48], PruneVid [21]), thus preventing a global understanding of video dynamics and comprehensive spatio-temporal optimization. Furthermore, despite the potential for synergy, no prior work has systematically explored integrating *inner-LLM* and *outer-LLM pruning* strategies or analyzed their mutual benefits. The current methods, while offering some benefits, still leave room for improvement.

Table 1: **Compression scope of vision-language model acceleration methods.** This table outlines where different methods apply compression. *Spatial* and *Temporal* refer to compression of the input visual data, while *Inner-LLM* indicates compression mechanisms applied within the model's processing.

| Methods | Spatial | Temporal | Inner-LLM |
|---|---|---|---|
| FastV [6] | ❌ | ❌ | ✅ |
| PDrop [61] | ❌ | ❌ | ✅ |
| LLaVA-PruMerge [42] | ✅ | ❌ | ❌ |
| VisionZip [65] | ✅ | ❌ | ❌ |
| DyCoke [48] | ❌ | ✅ | ✅ |
| FastVID [44] | ✅ | ✅ | ❌ |
| Ours | ✅ | ✅ | ✅ |

To address these limitations, we propose a holistic token pruning for video LLMs that leverages external and internal strategies. Our method first tackles temporal redundancy through a global redundancy-aware video segmentation process, followed by spatio-temporal merging. This external step reduces visual tokens to less than 10%, significantly alleviating the computational burden on the subsequent LLM. Complementing this, we introduce a new and robust inner-LLM token similarity-based merging method, specifically designed for integration with our outer-LLM pruning method, enabling mutual benefits. This integrated strategy offers a more holistic and efficient solution to handle long videos with LLMs, as summarized in Tab. 1, which contrasts the compression achieved by our approach in both the spatio-temporal domain and within the inner-LLM against other methods.

Empirical evaluations validate the effectiveness of our proposed method in achieving a compelling efficiency-performance trade-off. Specifically, as shown in Fig. 1 (right), on the LLaVA-OneVision-7B model [25], our approach reduces computational costs to just 6.9% of the original FLOPs while remarkably preserving 99.1% of the original model's performance. Moreover, we observe significant gains in inference efficiency, achieving a 2.28× reduction in Time-To-First-Token (TTFT) and a 1.32× acceleration in decoding throughput. These results clearly demonstrate the substantial practical advantages of our holistic token merging framework for efficient video LLM inference.

Our key contributions are summarized as follows:

1. We analyze the phenomenon of temporal redundancy in the context of video LLMs and propose a global redundancy-aware temporal merging method to effectively address the inefficiency in video LLMs before LLM processing in a plug-and-play fashion.

2. We introduce a robust inner-LLM similarity-based merging technique specifically designed for integration with the outer-LLM pruning method, facilitating synergistic optimization.

3. Extensive evaluations on LLaVA-OneVision and LLaVA-Video demonstrate that our integrated pruning framework achieves a state-of-the-art efficiency-performance trade-off, significantly reducing computational costs and accelerating inference while preserving model performance.

## 2 Related Work

### 2.1 Video Large Language Models

The rapid progress of multimodal large language models has led to the integration of video encoders, creating video LLMs that excel in video understanding and question answering tasks [63, 17, 25, 2, 56, 3, 27, 28, 30, 67, 47, 50, 23, 1, 46, 29]. However, the substantial number of tokens generated by processing numerous video frames hinders inference efficiency, thereby impeding the widespread adoption of video LLMs. Existing approaches have attempted to mitigate this issue. For instance, VideoLLaMA [67] employs a Q-Former module [26] to aggregate video tokens, while MovieChat [47] introduces a memory module to merge and store token representations. Although pooling mechanisms in LLaVA-OneVision [25] reduce token counts, each video frame still produces hundreds of tokens for downstream processing. Consequently, handling tens of thousands of visual tokens for long video inputs substantially increases inference time and memory consumption. While works such as VILA [32] and NVILA [35] aim to optimize token usage, these methods often require model fine-tuning, demanding considerable hardware resources [22, 30, 32, 35, 55, 33]. This underscores a critical need for developing more efficient, training-free token compression methods specifically for video LLMs, bypassing the need for costly model adaptations and significant hardware investment.

### 2.2 Visual Token Compression

Token compression [43] has emerged as an effective strategy for reducing token redundancy in vision transformers and large language models. ToMe [4] merges similar tokens in ViTs to alleviate spatial redundancy, while TempMe [45] focuses on minimizing temporal redundancy by merging adjacent video clips. TESTA [40] achieves up to a 75% reduction in processed tokens by employing temporal and spatial aggregation modules. For MLLMs, FastV [6] prunes non-essential visual tokens in early layers of LLM. TopV [64] proposes an optimization framework to prune unnecessary visual tokens. DyMU [57] introduces token merging in the visual encoder and virtual unmerging in the LLM decoder. PDrop [61] performs progressive pruning of tokens at different stages within the LLM. LLaVA-PruMerge [42] and VisionZip [65] leverage attention weight analysis in visual encoders to eliminate spatial redundancy. However, the inherent temporal dependencies between video frames necessitate specialized compression designs. Consequently, recent methods specifically for video token compression have gained increasing attention. DyCoke [48] consolidates tokens across frames and implements dynamic key-value cache reduction. PruneVID [21] clusters video tokens, whereas FastVID [44] enhances compression by combining temporal segmentation with spatio-temporal token merging. In this paper, we propose a new token merging strategy specifically designed for video LLMs, which fully considers spatio-temporal characteristics to maximize performance retention.

## 3 Method

### 3.1 Background on Video LLMs Inference

The inference process of video LLMs involves three key stages: before LLM, prefilling, and decoding.

**(1) Before LLM.** Given an input video with $B$ frames, a vision encoder processes each frame to produce $N_v$ embedding vectors. These are projected into the text embedding space, yielding visual tokens $H_v \in \mathbb{R}^{BN_v \times d}$, where $d$ represents the dimension of the hidden state space. A text prompt $T = \{t_i\}_{i=1}^{N_q}$ is tokenized and embedded into text tokens $H_q \in \mathbb{R}^{N_q \times d}$ similarly. Finally, the visual and text tokens are concatenated to form $H = \text{concat}[H_v, H_q]$, which serves as the LLM input.

**(2) Prefilling Stage.** During prefilling, each transformer layer $l$ of the LLM performs self-attention operations on the concatenated input $H$. It begins with linear transformations to compute query $\mathbf{Q}^l = H\mathbf{W}_Q^l$, key $\mathbf{K}^l = H\mathbf{W}_K^l$, and value $\mathbf{V}^l = H\mathbf{W}_V^l$, where $\mathbf{W}_Q^l, \mathbf{W}_K^l, \mathbf{W}_V^l \in \mathbb{R}^{d \times d}$ are learnable projection matrices. The resulting key-value pairs ($\mathbf{K}^l$ and $\mathbf{V}^l$) are then cached (KV cache) to enhance the efficiency of token generation during the subsequent decoding phase.

**(3) Decoding Stage.** The decoding stage generates tokens autoregressively, leveraging the KV cache. At each time step $t$, only the new token $h_t$ is processed to compute its key and value representations, avoiding recalculating attention weights over the entire history. The KV cache is updated by appending the new key-value pairs: $\mathbf{K} \leftarrow [\mathbf{K}, h_t \mathbf{W}_K], \mathbf{V} \leftarrow [\mathbf{V}, h_t \mathbf{W}_V]$. This caching mechanism substantially reduces the computational complexity of the generation process.

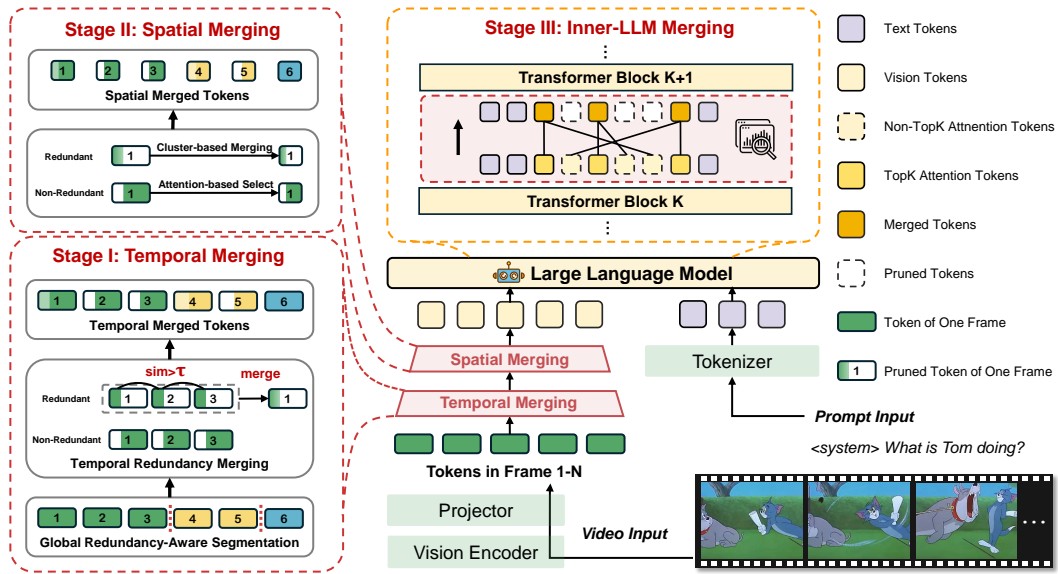

Figure 2: **Overview of our *HoliTom* method.** *HoliTom* compresses video LLMs across three scopes; the first two are outer-LLM pruning. **Temporal Merging** maximizes temporal compression via global redundancy-aware segmentation, merging similar tokens into their first occurrence. **Spatial Merging** further reduces redundancy by applying tailored spatial compression based on the characteristics of remaining temporal variations. **Inner-LLM Merging** leverages attention within the LLM to identify key tokens and merges less important, similar tokens, streamlining information within the LLM.

## 3.2  Global Redundancy-Aware Temporal Merging

Temporal redundancy describes feature persistence at fixed spatial locations across consecutive frames. We identify this redundancy for the $k$-th feature between frames $m$ and $m+1$ using their respective feature vectors $h_{m,k}$ and $h_{m+1,k}$. A feature is considered temporally redundant if its normalized similarity, $sim(h_{m,k}, h_{m+1,k})$, exceeds a defined threshold $\tau \in [0, 1]$.

For a temporal segment defined by a start frame $t_s$ and an end frame $t_e$ (covering $[t_s, t_e)$), the total number of prunable tokens, $g(t_s, t_e)$, is calculated. This involves counting tokens $N(t_s, t_e)$ that are consecutively redundant across *all* frames from $t_s$ to $t_e - 1$, and then multiplying by the number of subsequent frames $(t_e - t_s - 1)$ within the segment where these tokens can be pruned. Our method prunes the redundant by merging these subsequent occurrence tokens into their first appearance at start frame $t_s$, treating them as *temporal redundant tokens*, as shown in Fig. 2. The formulation is:

$$g(t_s, t_e) = \underbrace{\left( \sum_{k=1}^{N_v} \prod_{m=t_s}^{t_e-2} \mathbb{I}(sim(h_{m,k}, h_{m+1,k}) > \tau) \right)}_{N(t_s, t_e)} \times (t_e - t_s - 1), \tag{1}$$

where $N_v$ is the total number of features per frame, and $\mathbb{I}(\cdot)$ the indicator function.

Given a video of $B$ frames, our objective is to find a segmentation into $K$ consecutive segments $[t_i, t_{i+1})$ (with $t_1 = 1$, $t_{K+1} = B + 1$, and $t_i < t_{i+1}$) that maximizes the total prunable features:

$$\max_{K, \{t_i\}_{i=1}^{K+1}} \sum_{i=1}^{K} g(t_i, t_{i+1}). \tag{2}$$

This optimization is solved using dynamic programming to achieve *global* optimization. Let $dp[i]$ be the maximum prunable features for a video ending at frame $i$ (exclusive, i.e., considering frames $1, ..., i-1$), where frame $i$ marks the exclusive end of the last segment. The value $prev[i]$ stores the optimal starting frame $j^*$ of this final segment $[j^*, i)$. The state transition is given by:

$$dp[i] = \max_{1 \le j < i}\{dp[j] + g(j, i)\}, \quad \text{with} \quad prev[i] = \arg\max_{1 \le j < i}\{dp[j] + g(j, i)\}. \tag{3}$$

The base case is $dp[1] = 0$. The maximum prunable features for the entire video are $dp[B + 1]$. The optimal segmentation is reconstructed by backtracking from $B + 1$ using the $prev$ array.

## 3.3 Spatial Merging

After temporal merging, tokens are classified as *non-redundant* or *redundant* temporal tokens. We first process the former. Inspired by works [42, 65, 69, 52], we utilize the CLS tokens for spatial feature selection. For vision encoders like Siglip [66] that do not have an explicit CLS token, a method is detailed to derive CLS-equivalent attention. Specifically, we compute the attention matrix:

$$A = \text{Softmax}(QK^T/\sqrt{d}) \in \mathbb{R}^{B \times N_v \times N_v}, \tag{4}$$

where $d$ is the state dimension. Token importance is quantified by averaging the attention weights each token receives from all other tokens within the same frame in the vision tower, yielding a score vector $A_{\text{avg}} \in \mathbb{R}^{B \times N_v}$. Tokens receiving higher average attention are considered more salient.

Consistent with video LLMs of applying spatial pooling (e.g., after the projector to reduce tokens), we reshape $A_{\text{avg}}$ to its original spatial grid dimensions ($H \times W = N_v$) and apply an analogous pooling operation. This results in a spatially downsampled importance map $\overline{A}_{\text{avg}} \in \mathbb{R}^{B \times \overline{H} \times \overline{W}}$. Ultimately, we select the visual features corresponding to the highest scores in $\overline{A}_{\text{avg}}$ as the representative and most informative spatial tokens, known as *attention-based select*, discarding all others.

The computation of vision tower attention is intra-frame. Averaging these attention weights across frames lacks theoretical justification, invalidating the attention-based method for *redundant temporal tokens*. To process these features, we employ a *cluster-based merging* method utilizing density peak clustering based on $k$-nearest neighbors (DPC-KNN) [13, 41]. Given a set of $N$ redundant temporal tokens $[v_1, v_2, ..., v_N]$ within the first frame of the segmentation. For each token $v_i$, we calculate its local density $\rho_i$, distance to the closest higher-density token $\delta_i$ and the final density score $\gamma_i = \rho_i \times \delta_i$:

$$\rho_i = \exp\left(-\frac{1}{k} \sum_{v_j \in \text{kNN}(v_i)} d(v_i, v_j)^2\right), \quad \delta_i = \begin{cases} \max_{j \neq i} d(v_i, v_j) & \text{if } \rho_i = \max_k \rho_k \\ \min_{j:\rho_j > \rho_i} d(v_i, v_j) & \text{otherwise} \end{cases}. \tag{5}$$

Tokens with high $\gamma_i$ are selected as cluster centers. After selecting the cluster centers, each remaining feature is assigned to the cluster whose center is closest in feature space. Finally, the representative feature for each cluster is then computed by averaging the features assigned to it. Ultimately, the compressed features, derived from this clustering process, along with the non-redundant features, are concatenated according to their original spatial order, thereby preserving positional characteristics.

## 3.4 Inner-LLM Merging

Inefficient visual attention in large vision language models has been widely discussed [6, 61]. Existing methods, such as FastV [6] and PDrop [61], directly *discard* redundant visual tokens, which may lead to performance degradation due to information loss. Unlike these approaches, our proposed method addresses it by *merging* the information from potentially redundant tokens instead of simply discarding them. Specifically, at the K-th layer of the LLM, to reduce the number of visual tokens by R%, we employ a token selection strategy based on attention scores. We use the attention weights of the last token to rank all vision tokens at layer K. The R% of visual tokens exhibiting the lowest attention scores are identified as candidates for merging. We find its most similar visual token within the set of tokens designated for retention. For a retained token $v_r$ and its associated set of low attention tokens $V_m = \{v_{m_1}, v_{m_2}, ..., v_{m_n}\}$. The updated retained token $v_r'$ is:

$$v_r' = \text{average}(v_r, v_{m_1}, ..., v_{m_n}). \tag{6}$$

This selective merging preserves relevant features from tokens that would otherwise be removed, mitigating information loss while achieving the desired token reduction.

# 4 Experimental Results

## 4.1 Experimental Settings

**Benchmarks.** We evaluate our method on four widely-used video understanding benchmarks: MVBench [28], EgoSchema [37], LongVideoBench [59], and VideoMME [17]. Comprising videos of varying lengths and complex scenarios, these benchmarks provide a comprehensive testbed for assessing the effectiveness and generalization of our method.

Table 2: **Comparison of state-of-the-art methods across benchmarks. Best** and **most efficient** results are in bold, second best underlined. Here, "HoliTom" means the full version of our method; "HoliTom (w/o M)" means our method *without* inner-LLM merging, for reference.

| Method | Prefilling FLOPs (T) ↓ | FLOPs Ratio ↓ | Before LLM Retained Ratio | MVBench ↑ | EgoSchema ↑ | LongVideo Bench ↑ | VideoMME ↑ | Avg. ↑ Score | Avg. ↑ % |
|---|---|---|---|---|---|---|---|---|---|
| LLaVA-OV-7B | 40.8 | 100% | 100% | 58.3 | 60.4 | 56.4 | 58.6 | 58.4 | 100 |
| FastV [6] | 9.3 | 22.8% | 100% | 55.9 | 57.5 | 56.7 | 56.1 | 56.5 | 96.7 |
| PDrop [61] | 10.5 | 25.7% | 100% | 56.1 | 58.0 | 54.1 | 56.4 | 56.2 | 96.2 |
| DyCoke [48] | 8.7 | 21.3% | 25% | 53.1 | 59.5 | 49.5 | 54.3 | 54.1 | 92.6 |
| VisionZip [65] | 8.7 | 21.3% | 25% | 57.9 | 60.3 | 56.5 | 58.2 | 58.2 | 99.7 |
| PruneVid [21] | 8.7 | 21.3% | 25% | 57.4 | 59.9 | 55.7 | 57.4 | 57.6 | 98.6 |
| FastVID [44] | 8.7 | 21.3% | 25% | 56.5 | - | 56.3 | 58.0 | - | - |
| **HoliTom (w/o M)** | 8.7 | 21.3% | 25% | **58.5** | 60.8 | 56.5 | **59.1** | 58.7 | 100.5 |
| **HoliTom** | **7.1** | **17.4%** | 25% | 58.4 | **61.2** | **56.7** | 58.9 | **58.8** | **100.7** |
| VisionZip [65] | 7.0 | 17.2% | 20% | 57.7 | 59.8 | 55.2 | 57.9 | 57.7 | 98.8 |
| PruneVid [21] | 7.0 | 17.2% | 20% | 57.2 | 59.7 | 54.7 | 56.9 | 57.1 | 97.8 |
| FastVID [44] | 7.0 | 17.2% | 20% | 56.3 | - | **57.1** | 57.9 | - | - |
| **HoliTom (w/o M)** | 7.0 | 17.2% | 20% | 58.5 | 60.7 | 56.3 | **58.6** | 58.5 | 100.2 |
| **HoliTom** | **5.8** | **14.2%** | 20% | **58.7** | **61.0** | **57.1** | **58.6** | **58.8** | **100.7** |
| VisionZip [65] | 5.2 | 12.7% | 15% | 56.5 | 59.8 | 54.4 | 56.1 | 56.7 | 97.1 |
| PruneVid [21] | 5.2 | 12.7% | 15% | 56.8 | 59.7 | 55.4 | 56.6 | 57.1 | 97.8 |
| FastVID [44] | 5.2 | 12.7% | 15% | 56.0 | - | 56.2 | 57.7 | - | - |
| **HoliTom (w/o M)** | 5.2 | 12.7% | 15% | **58.1** | 61.0 | **57.0** | **58.1** | **58.5** | **100.2** |
| **HoliTom** | **4.3** | **10.5%** | 15% | **58.1** | **61.2** | 56.4 | 57.3 | 58.2 | 99.7 |
| VisionZip [65] | 3.4 | 8.3% | 10% | 53.5 | 58.0 | 49.3 | 53.4 | 53.5 | 91.6 |
| PruneVid [21] | 3.4 | 8.3% | 10% | 56.2 | 59.8 | 54.5 | 56.0 | 56.6 | 96.9 |
| FastVID [44] | 3.4 | 8.3% | 10% | 55.9 | - | 56.3 | **57.3** | - | - |
| **HoliTom (w/o M)** | 3.4 | 8.3% | 10% | **56.9** | 61.1 | **56.5** | 56.9 | 57.8 | 99.0 |
| **HoliTom** | **2.8** | **6.9%** | 10% | **57.3** | **61.2** | 56.3 | 56.8 | **57.9** | **99.1** |

**Compared Methods.** We compare our proposed *HoliTom* against **6** strong training-free baselines: 1) FastV [6], identifies key tokens during prefilling using attention scores between predicted and vision tokens; 2) PDrop [61], prunes visual tokens within partitioned LLM stages, guided by image and instruction tokens; 3) Visionzip [65], prunes tokens before LLM via spatial token merging; 4) DyCoke [48], employs temporal merging before LLM and dynamic KV cache pruning in decoding; 5) PruneVid [21], minimizes video redundancy via spatio-temporal token clustering; and 6) FastVID [44], a concurrent work, partitions videos and applies density-based token pruning. Due to the lack of public code, we compare FastVID to its reported results. For all other baselines and our method, experiments use their open-source code under identical hardware condition.

**Inference Cost Evaluation.** We evaluate the inference cost of transformer layers, each composed of multi-head attention (MHA) and feed-forward network (FFN) modules. Following previous work [6, 61, 48], the FLOPs for processing $n_i$ *vision* tokens in layer $i$, with hidden state size $d$ and FFN intermediate size $m$, are defined as $4n_id^2 + 2n_i^2d + 2n_idm$. For an LLM with $T$ transformer layers, the total FLOPs span the prefilling and decoding phases, calculated as:

$$\sum_{i=1}^{T} \underbrace{(4n_id^2 + 2n_i^2d + 2n_idm)}_{\text{Prefilling FLOPs per layer}} + \underbrace{R((4d^2 + 2dm) + 2(dn_i + \frac{1}{2}d(R+1)))}_{\text{Decoding FLOPs per layer}}. \tag{7}$$

For consistency, the decoding calculation is fixed for predicting $R = 100$ tokens, accounting for the the KV cache. In video LLMs, the decoding phase FLOPs contribute only approximately **2%** of the total. Consequently, our primary optimization focus is on the *prefilling* stage. When considering prefilling optimization, inner-LLM pruning methods like FastV [6], the FLOPs incurred in the first 2 shallow layers can amount to 2.9 TFLOPs in LLaVA-OneVision-7B. Even pruning 100% token in the layer, these methods cannot match the potential efficiency compared to outer-LLM pruning methods. Thus, outer-LLM pruning offers a more impactful optimization approach for this domain.

**Implementation Details.** Our method is implemented on LLaVA-OneVision-7B/72B [25] and LLaVA-Video-7B [70] models. Evaluation uses NVIDIA A100 GPUs, while inference is on an RTX A6000. Inference cost is measured by prefilling FLOPs, with baselines configured for comparable FLOPs (details in the appendix A). The default $\tau$ is 0.8; for 10% compression, $\tau$ is 0.65. Following official practice, LLaVA-OneVision models utilize 32 input video frames ($N_v = 196$), while LLaVA-Video uses 64 frames ($N_v = 169$). All benchmarks are conducted using LMMs-Eval [68, 24].

Table 3: **Cross-backbone method comparison.** Performance comparison of our method against state-of-the-art methods across different backbones, demonstrating consistent effectiveness.

| Model | Method | Prefilling FLOPs (T)↓ | FLOPs Ratio↓ | Before LLM Retained Ratio | MVBench↑ | EgoSchema↑ | LongVideo Bench↑ | VideoMME↑ | Avg.↑ Score | Avg.↑ % |
|---|---|---|---|---|---|---|---|---|---|---|
| LLaVA-OneVision-72B | Vanilla | 429.3 | 100% | 100% | 60.9 | 61.1 | 62.7 | 65.7 | 62.6 | 100 |
| | FastV [6] | 86.4 | 20.1% | 100% | 56.1 | 57.1 | 57.0 | 61.2 | 57.9 | 92.5 |
| | Visionzip [65] | 59.0 | 13.7% | 15% | 58.4 | 59.3 | 57.4 | 63.8 | 59.7 | 95.4 |
| | PruneVid [21] | 59.0 | 13.7% | 15% | 56.8 | 57.7 | 57.4 | 62.7 | 58.6 | 93.6 |
| | **HoliTom (w/o M)** | 59.0 | 13.7% | 15% | 58.5 | 60.0 | 57.8 | 64.1 | 60.1 | 96.0 |
| | **HoliTom** | **51.6** | **12.0%** | 15% | **58.7** | **60.1** | 57.2 | **64.3** | 60.1 | 96.0 |
| LLaVA-Video-7B | Vanilla | 80.2 | 100% | 100% | 60.4 | 57.2 | 58.9 | 64.3 | 60.2 | 100 |
| | FastV [6] | 17.1 | 21.3% | 100% | 54.3 | 54.1 | 55.0 | 58.8 | 55.6 | 92.4 |
| | PDrop [61] | 19.5 | 24.3% | 100% | 55.9 | 54.3 | 54.7 | 61.9 | 56.7 | 94.2 |
| | VisionZip [65] | 9.3 | 11.6% | 15% | 56.7 | 54.7 | 54.7 | 60.7 | 56.7 | 94.2 |
| | **HoliTom (w/o M)** | 9.3 | 11.6% | 15% | **57.8** | 54.8 | 55.6 | 61.9 | 57.5 | 95.5 |
| | **HoliTom** | **7.6** | **9.5%** | 15% | 57.7 | 54.8 | 56.2 | 62.1 | 57.7 | 95.8 |

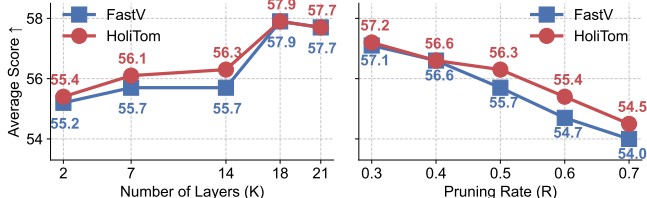

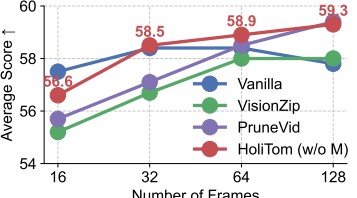

Figure 3: **Left:** Performance of our method *vs.* FastV when pruning various layers at rate R=50%. **Right:** Performance comparison with varying pruning rates at a fixed layer (K=14).

Figure 4: Performance *vs.* number of frames for our method and other token compression methods.

## 4.2 Main Results

**Comparison with State-of-the-Art Methods.** Tab. 2 benchmarks *Holitom* against state-of-the-art approaches on the LLaVA-OneVision-7B model, analyzing performance and inference cost (FLOPs) at various token retention ratios (25%, 20%, 15%, and 10%) prior to LLM processing. Inner-LLM pruning methods, such as FastV [6] and PDrop [61], often struggle to balance performance and efficiency, especially at lower token retention ratios (25%). DyCoke [48], which segments video frames into groups of 4 and prunes all but the first frame, is limited by its design, capping its lowest retention ratio at 25%. Spatial pruning methods like VisionZip [65] show a significant performance drop (up to 8.4%) at 10% retention. This decline stems from relying solely on spatial compression, less effective at preserving crucial temporal information needed for performance under aggressive pruning. Crucially, even **without** our inner-LLM merging technique, our method achieves state-of-the-art performance and efficiency consistently across the evaluated retention ratios. This highlights the superior robustness and adaptability of our approach compared to prior methods. Our inner-LLM merging method further enhances efficiency, driving optimization further. For instance, we retain only 6.9% of the original FLOPs, while preserving 99.1% of the baseline performance.

**Performance Comparison Across Different Backbones.** Tab. 3 assesses our method's performance across various backbones. For the powerful LLaVA-OneVision-72B model, sensitive to aggressive compression, our approach reduces computational cost to 11.3%, keeping 96% of its original performance. LLaVA-Video-7B presents a greater compression challenge due to its higher initial pooling rate (169 *vs.* 196 tokens/frame in LLaVA-OneVision). Despite this, our method achieves a reduction to just 9.5% of the original FLOPs, retaining 95.8% performance and outperforming existing methods. Overall, achieving significant token compression with minimal performance drop is indeed tougher for LLaVA-OneVision-72B and LLaVA-Video-7B than for LLaVA-OV-7B.

**HoliTom *vs.* FastV under Outer-LLM Compression.** Building on the challenges faced by inner-LLM pruning methods discussed in Section 4.1, we compare our inner-LLM merging method with FastV, specifically in scenarios where outer-LLM compression is already applied. In this compressed context, the property "*an image is worth 1/2 tokens after layer 2*" [6] is not consistently observed. This is because outer-LLM compression concentrates information, making trivial token discarding more difficult using attention mechanisms. As illustrated in Fig. 3, our method demonstrates superior performance compared to FastV when pruning 50% at shallower layers. Furthermore, at equivalent layers, our approach consistently surpasses FastV across a wide range of pruning rates, underscoring its effectiveness. This effectiveness stems from our inner-LLM merging method, which better preserves information rather than directly discarding it.

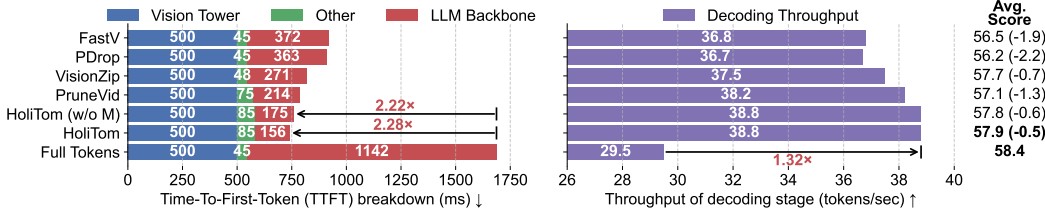

Figure 5: **Achieving superior inference.** "Other" indicates token pre-processing time (e.g., pooling). Our proposed method reduces Time-To-First-Token (TTFT) by 2.28× and achieves 1.32× higher decoding throughput, outperforming all other token compression methods and the vanilla model.

Table 4: **Ablation study on merging modules.** Our temporal merging module reduces FLOPs to 75.7% without performance loss, alleviates the performance degradation caused by aggressive spatial pruning. The integration of all 3 modules achieves the best performance-efficiency trade-off.

| Method | Prefilling FLOPs (T) ↓ | FLOPs Ratio ↓ | Before LLM Retained Ratio | MVBench ↑ | EgoSchema ↑ | LongVideo Bench ↑ | VideoMME ↑ | Avg. ↑ Score | % |
|---|---|---|---|---|---|---|---|---|---|
| Vanilla | 40.8 | 100% | 100% | 58.3 | 60.4 | 56.4 | 58.6 | 58.4 | 100 |
| Only Temporal | 30.9 | 75.7% | 79% | **58.9** | 60.5 | 56.5 | **59.1** | **58.8** | **100.7** |
| Only Spatial | 5.2 | 12.7% | 15% | 57.9 | 60.8 | 54.2 | 56.8 | 57.4 | 98.3 |
| **HoliTom (w/o M)** | 5.2 | 12.7% | 15% | 58.1 | 61.0 | **57.0** | 58.1 | 58.5 | 100.2 |
| **HoliTom** | **4.3** | **10.5%** | 15% | 58.1 | **61.2** | 56.4 | 57.3 | 58.2 | 99.7 |

**Performance Scaling with more frames.** Our method scales performance robustly with increasing input frames (Fig. 4). A challenge for video LLMs is that uniformly sampled frames may miss crucial information required for accurate answers. Therefore, an effective token pruning method is essential to process more frames and capture sufficient context. Fig. 4 shows our approach consistently outperforms other compression methods across frame rates. At 16 frames, where less temporal redundancy exists, our method, while slightly below the vanilla, still outperforms all other compression techniques. With 64 frames, our method is more efficient and achieves superior performance over the vanilla model. Furthermore, when processing 128 frames, our token compression approach avoids the maximum context length limitations that bottleneck vanilla models. This capability is particularly beneficial for tasks that require an extensive temporal context or to answer complex questions with long text, resulting in improved performance.

**Discussion: Improved Performance after Token Compression** Tabs. 2, 4, and Fig. 4 present a key finding: models employing our token compression technique **outperform the original models** on various benchmarks. This surprising result underscores a fundamental principle for achieving superior performance at the input stage: *the value of key information over exhaustive information*. Excessive, irrelevant, or redundant data acts as noise, obscuring essential signals critical for effective processing. This information overload impedes the capacity of the model to accurately identify and process critical details, thereby degrading understanding and response generation. By providing a refined input that retains pertinent information while shedding redundant information, our compression method facilitates deeper comprehension and yields more accurate, relevant outputs. Collectively, these results underscore the efficacy of our technique in distilling key information and demonstrate that intelligent input refinement is crucial for superior model performance.

## 4.3 Efficiency Results

Fig. 5 summarizes the impact of various token compression methods on the inference efficiency of video LLMs. As shown, all the evaluated methods demonstrably reduce LLM prefilling time. Our method, in particular, reduces it to just 13.7% of the original. For VisionZip [65], PruneVid [21], and our *HoliTom* require token pre-processing, which introduces additional "other" time. Furthermore, both PruneVid and our method produce a variable number of tokens per frame, complicating batch processing, which contributes to extra overhead. Our method, designed to maximize temporal redundancy pruning, leads to finer-grained segmentation, further influencing this observation. Nevertheless, our method achieves the maximum reduction in Time-To-First-Token latency, reducing it by 2.28×, while maintaining optimal performance. Although we did not specifically optimize for decoding, our model's decoding speed still benefits from the reduced number of vision tokens. Our throughput increased by 1.32× compared to the original model, the highest among all methods evaluated.

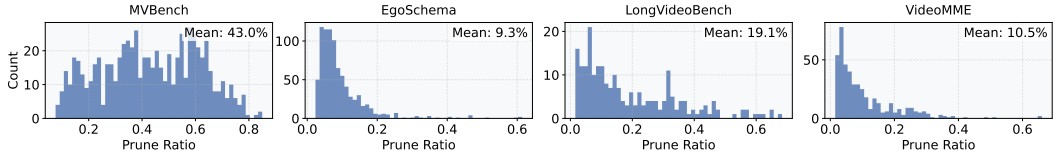

Figure 6: **Histogram of temporal pruning rates across four benchmarks** ($\tau = 0.80$). The average pruning ratio for each benchmark is annotated in the top right. MVBench (16s duration) exhibits the highest ratio, reflecting greater temporal redundancy, while EgoSchema is the least.

Table 5: **Ablation study on video segmentation methods.** This table compares different video segmentation strategies: Fixed-interval segmentation partitions the video at equal intervals; DySeg adaptively segments based on transition similarity; and our proposed global redundancy-aware segmentation.

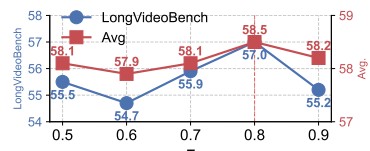

| Methods | MVBench | EgoSchema | LongVideo Bench | VideoMME | Avg. |
|---|---|---|---|---|---|
| Fixed-interval | **57.0** | 60.9 | 53.8 | 56.4 | 57.0 |
| DySeg [44] | 56.8 | 60.8 | 54.1 | 56.6 | 57.1 |
| HoliTom (w/o M) | 56.9 | **61.1** | **56.5** | **56.9** | **57.8** |

Figure 7: **Ablation study on $\tau$.** Performance of our method is analyzed with varying $\tau$ at a target before LLM retained ratio of 15%.

## 4.4 Ablation Study

**Ablation study on merging modules.** Tab. 4 provides a detailed ablation study on the contribution of our proposed merging modules. We first evaluated the temporal merging module ($\tau = 0.8$), designed to eliminate temporal redundancy, which demonstrated efficiency gains while preserving performance. Across the four benchmarks, our method achieved 100.7% of the baseline performance while reducing FLOPs to 75.7%. Note that the reported average pruning rate is calculated over four datasets. The average pruning rate varied across datasets is illustrated in Fig. 6. For instance, MVBench(16s), with its shortest duration, exhibits the highest temporal redundancy, allowing approximately 43% pruning, whereas EgoSchema contains the least, permitting only about 9.3%. We then investigate combining temporal with spatial pruning. Applying our temporal pruning method significantly mitigates the performance degradation typically associated with aggressive spatial pruning alone. Furthermore, incorporating the inner merging module allowed us to push the efficiency boundaries even further, ultimately retaining 99.7% performance with a mere 10.5% of the original FLOPs.

**Ablation study on temporal segmentation method.** Tab. 5 compares different temporal segmentation methods. Fixed-interval Segmentation generates 8 segments with an interval of 4. DySeg [44] selects segment start points using the 8 largest inter-frame differences and includes frames below a 0.90 similarity threshold. Our proposed global redundancy-aware segmentation maximally leverages spatial redundancy and achieves a better performance.

**Ablation study on $\tau$.** The hyperparameter $\tau$ controls the sensitivity of the temporal pruning mechanism, with lower values leading to more aggressive pruning. For a fixed retained ratio, $\tau$ also governs the balance between the amount of spatial and temporal pruning applied. Fig. 7, demonstrates the easy tunability of $\tau$, with peak performance observed around $\tau = 0.8$. This value is adopted uniformly without performance degradation, as shown in Tab. 4. For a 10% pruning target, we set $\tau = 0.65$ to mitigate performance degradation from aggressive spatial pruning.

## 5 Conclusion

This paper presents *HoliTom*, a new training-free holistic token merging framework for boosting the efficiency of video LLMs by effectively handling redundant visual tokens. *HoliTom* achieves this through a synergistic integration of outer-LLM spatio-temporal reduction, drastically reducing initial token counts, and a robust inner-LLM token merging mechanism tailored for compatibility and further optimization. Evaluated on prominent video LLMs, *HoliTom* achieves a state-of-the-art efficiency-performance trade-off, substantially reducing computational costs (e.g., to 6.9% FLOPs) while preserving high performance (e.g., 99.1% accuracy), and accelerating inference (2.28× TTFT, 1.32× throughput). These results underscore the effectiveness of *HoliTom* in enabling practical and efficient video LLMs inference for complex, long-form video understanding.

## Acknowledgment

This paper is supported by Young Scientists Fund of the National Natural Science Foundation of China (No. 62506305), Zhejiang Leading Innovative and Entrepreneur Team Introduction Program (No. 2024R01007), Key Research and Development Program of Zhejiang Province (No. 2025C01026), Scientific Research Project of Westlake University (No. WU2025WF003), Chinese Association for Artificial Intelligence (CAAI) & Ant Group Research Fund - AGI Track (No. 2025CAAI-ANT-13), and the Special Support Talents Program of "Xi Hu Ming Zhu Program" in Hangzhou.

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

Table 6: **Comparison of state-of-the-art methods on Qwen2.5-VL 7B.** Qwen2.5-VL accepts video input at 2 fps, capped at a maximum of 768 frames, maximum video token limit to 24,576 (as the technical report recommends). A/B in the # Token column indicates that A tokens are first provided to the LLM, and then compressed to B tokens during the LLM forward pass. The token number is derived from the average video token count per task within VideoMME. FastV performs full attention matrix calculation in memory, causing OOM errors due to the large number of video tokens.

| Method | # Token ↓ | | TFLOPs ↓ | | VideoMME ↑ | | | | |
|---|---|---|---|---|---|---|---|---|---|
| | | | | | Short | Medium | Long | Overall | |
| Qwen2.5-VL-7B | 18442 | 100% | 377.2 | 100% | 77.4 | 68.1 | 55.6 | 67.0 | 100% |
| FastV [6] | 18442 / 9221 | 100% / 50.0% | 170.4 | 45.2% | | | OOM | | |
| Visionzip [65] | 9221 | 50.0% | 154.5 | 41.0% | **74.9** | **66.6** | 55.7 | 65.7 | 98.1% |
| PruneVid [21] | 9173 | 49.7% | 153.5 | 40.7% | 72.3 | 64.8 | 54.7 | 63.9 | 95.4% |
| HoliTom (w/o M) | **6513** | **34.9%** | **102.0** | **27.0%** | 74.4 | 66.4 | **56.4** | 65.8 | **98.2%** |
| FastV [6] | 18442 / 4610 | 100% / 25.0% | 90.7 | 24.0% | | | OOM | | |
| Visionzip [65] | 4610 | 25.0% | 68.7 | 18.2% | **73.1** | 63.3 | 55.9 | 64.1 | 95.7% |
| PruneVid [21] | 4632 | 25.1% | 69.1 | 18.3% | 69.3 | 61.1 | 53.2 | 61.2 | 91.3% |
| HoliTom (w/o M) | **4504** | **24.4%** | **66.9** | **17.7%** | 72.7 | **65.7** | **56.1** | 64.8 | **96.7%** |

# A    Supplemental Implementation Details

Our method is implemented on the LLaVA-OneVision-7B/72B [25] and LLaVA-Video-7B [70] models. Evaluation utilized NVIDIA A100 (80GB) GPUs; inference was performed on an NVIDIA RTX A6000 GPU. To ensure a fair comparison of computational cost, we used total prefilling FLOPs as the primary metric. Baselines are configured for comparable FLOPs: FastV [6] prunes 80% of tokens at layer 2; PDrop [61] retains 50%, 25%, and 12.5% of vision tokens at layers 2, 7, and 14, respectively; VisionZip [65] and PruneVid [21] maintain a consistent proportion of input tokens with our method. Performance results for FastVID [44] are adopted directly from their original paper. For our proposed method, the default threshold $\tau$ is 0.8. In the specific experiments conducted on Qwen2.5-VL [3] with a maximum sampling of 768 frames, a lower threshold of $\tau = 0.2$ was used. This adjustment accounts for the higher temporal redundancy present when sampling is dense. In experiments targeting a 10% compression ratio, $\tau$ was set to 0.65. Experimental setups include pruning K=18 layers of the 7B model and K=60 layers of the 72B model, both at a ratio of R=50%. Following the official LLaVA-OneVision specifications, the default input video frames are 32 and $N_v = 196$. For LLaVA-Video, the default input consisted of 64 video frames with $N_v = 169$. All benchmark evaluations are performed using the LMMs-Eval [68, 24].

# B    Supplemental Experimental Results

## B.1    Experiments on Qwen2.5-VL with High Frame Sampling

Existing models like LLaVA-OV [25] and LLaVA-Video [70] utilize a fixed input of 32/64 video frames, each resized to a static resolution (Tab. 2, 3). In contrast, frontier models, such as Qwen2.5-VL [3], introduce advanced features including FPS frame sampling, which extend input sequences (up to 768 frames), and dynamic resolution support. These new capabilities pose new challenges to existing token compression methods in maintaining performance for video understanding tasks.

As shown in Tab. 6, HoliTom surpasses state-of-the-art methods across both token compression rates, especially for long videos. Due to Out-of-Memory (OOM) issues arising from the full attention matrix calculation, we were unable to report results for FastV and the inner-LLM merging execution.

## B.2    Impact of Token Compression on Fine-Grained Object Understanding

HoliTom applies aggressive video token compression. Does this aggressive compression impair the model's ability to comprehend fine-grained details? Tab. 7 presents the performance results on selected subtasks from the MVBench benchmark (originally detailed in Table 2).

Table 7: **Fine-grained object tasks.** Performance on MVBench object subtasks (object existence (OE), object interaction (OI), and object shuffle (OS)) improves with more aggressive token compression.

| Method | FLOPs (%) | MVBench | | | |
| --- | --- | --- | --- | --- | --- |
| | | OE | OI | OS | Overall |
| OV 7B | 100 | 57.5 | 84.0 | 35.5 | 58.3 |
| +HoliTom | 17.4 | 61.0 | 83.5 | 36.5 | 58.4 |
| +HoliTom | 14.2 | **63.0** | 84.0 | 37.5 | **58.7** |
| +HoliTom | 10.5 | 60.5 | **84.5** | **38.0** | 58.1 |

Table 8: **Improved performance with enhanced efficiency.** Using more input frames, token compression boosts performance while controlling computational overhead.

| Method | # Frame | FLOPs (%) | Avg. Score |
| --- | --- | --- | --- |
| OV 7B | 32 | 100 | 58.4 |
| +HoliTom | 32 | 12.7 | 58.5 |
| +HoliTom | 64 | 26.5 | 58.9 |
| +HoliTom | 128 | 56.6 | 59.3 |

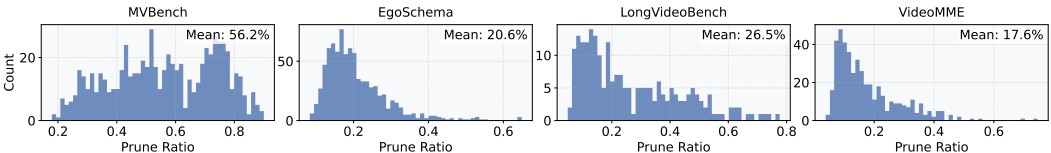

Figure 8: **Histogram of temporal pruning rates across four benchmarks** ($\tau = 0.65$). The average pruning ratio for each benchmark is annotated in the top right. MVBench (16s duration) exhibits the highest ratio, reflecting greater temporal redundancy, while VideoMME is the least ($\tau = 0.65$).

At compression rates ranging from 10% to 25%, HoliTom demonstrates increases in performance. This result would be improbable if HoliTom discards critical information about small objects. Instead, this outcome provides evidence of HoliTom's robust ability to retain fine-grained details.

## B.3 Enhanced Performance with Reduced Overhead

Tab. 8 extends the findings presented in Fig. 4. By sampling more frames with HoliTom while maintaining constant or even reduced total FLOPs, we achieve better performance compared to a vanilla model operating on fewer frames. We also observe that as the number of input frames increases, the computational overhead contributed by the vision encoder becomes a non-negligible factor. This performance and efficiency trade-off is further illustrated in Fig. 5 of our paper.

## B.4 Supplemental Ablation Study on $\tau$

In section 4.4, we discussed the selection of $\tau$ ($\tau = 0.8$) and the corresponding histogram of temporal pruning rates on four benchmarks for a retain ratio of 15%. Next, we detail the selection of the hyperparameter $\tau$ for a 10% retain ratio and present the corresponding histogram. As illustrated in Fig. 9, peak performance is observed around $\tau = 0.65$. The Fig. 8 presents the histogram of temporal pruning rates on the four datasets when $\tau = 0.65$. It is evident that controlling $\tau$ regulates the aggressiveness of temporal pruning; a larger $\tau$ results in more aggressive pruning.

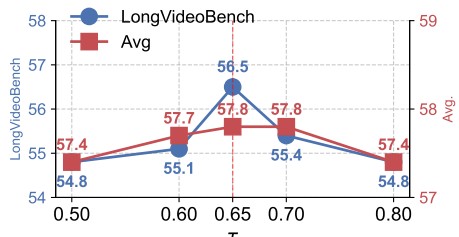

Figure 9: **Ablation study on $\tau$.** Performance of our method is analyzed with varying $\tau$ at a target before LLM retained ratio of 10%.

## B.5 Ablation Study on Merge Strategy

The core of spatio-temporal merging in HoliTom is a hybrid strategy that integrates attention-guided compression with similarity-guided clustering (DPC-KNN). As shown in Tab. 9, our mixed strategy achieves the best performance. The key to this lies in distinguishing between the two token types encountered during the merging process. For non-redundant tokens (within a single frame), the encoder's self-attention scores are *excellent priors*, making an attention-guided compression strategy highly effective. However, for redundant tokens (formed by merging tokens from adjacent frames), the original single-frame self-attention scores lack a theoretical basis as a merging metric. Therefore,

Table 9: **Ablation Study on Merge Strategy.** Our mixed strategy achieves the best performance.

| Method | MVBench | EgoSchema | LongVideoBench | VideoMME | Avg. Score |
|--------|---------|-----------|----------------|----------|------------|
| Attention | **58.4** | **60.9** | 55.9 | 57.4 | 58.1 |
| DPC-KNN | 57.4 | 59.6 | 53.9 | 56.6 | 56.9 |
| HoliTom | **58.4** | **60.9** | **56.2** | **58.3** | **58.5** |

DPC-KNN clustering is adopted to group these merged tokens based on feature similarity, which provides a more principled and effective approach in this specific spatio-temporal context.

## C    Compatible with Flash Attention

Our approach introduces two distinct merging strategies: inner-LLM and outer-LLM. The inner-LLM strategy, similar to prior work [6, 61, 48], is designed for integration with highly optimized attention implementations (e.g., Flash Attention [12, 11]). This requires obtaining attention scores from a specific layer *only once* during the prefilling stage, an operation introducing negligible computational overhead compared to total inference cost. In contrast, our outer-LLM merging strategy operates externally to the model, decoupled from the attention mechanisms of LLM.

## D    Limitations and Future Work

While our work demonstrates an adequate acceleration of video LLMs by token merging for inference, it is important to outline its current limitations. First, the approach is primarily designed for fixed-length video clips and does not natively support online, arbitrary-length streaming video input. This poses challenges for real-time processing [5, 39, 38] and maintaining long-term context understanding. Second, as shown in Fig. 5, similar to other methods [6, 61, 65, 48] in the token pruning area, our approach does not optimize the latency of the vision tower. Further work, such as quantization [31, 60, 16, 20], methods to accelerate the vision tower [10, 53, 57] and new application areas [58, 54, 62, 49, 7, 15, 14], is worth exploring for further optimization.

## E    Broader impacts

This work significantly enhances video LLM efficiency, addressing a key barrier to deployment and scalability. By reducing computational needs, it broadens access to advanced video AI, enabling wider application and fostering innovation.

## F    More Visualizations

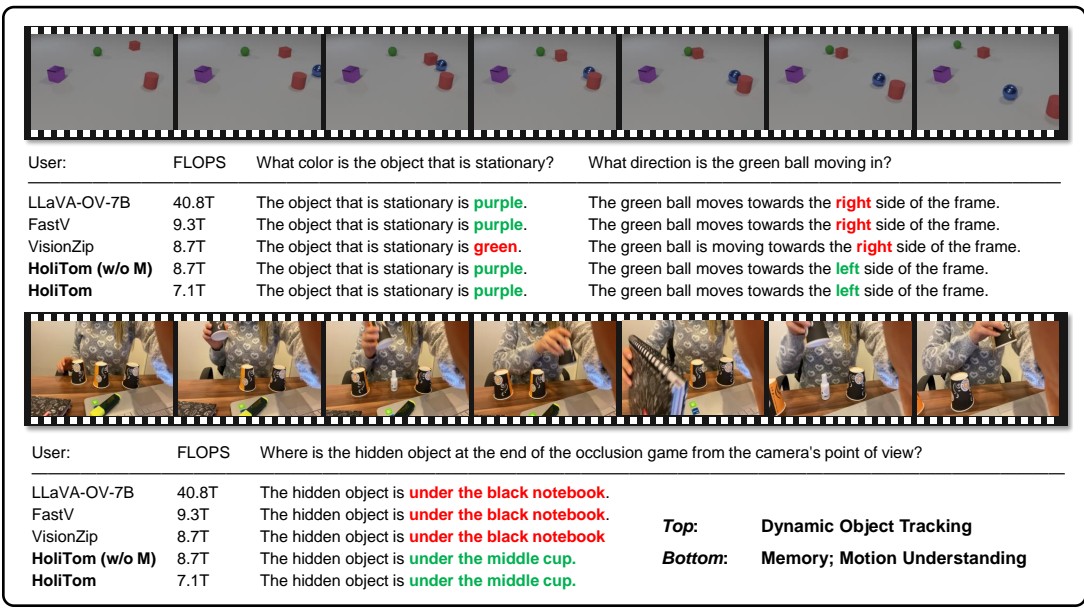

Figure 10: **Comparison on Challenging Video Understanding.** Green: correct results, Red: incorrect results. Our method is able to produce correct answers on challenging video tasks.

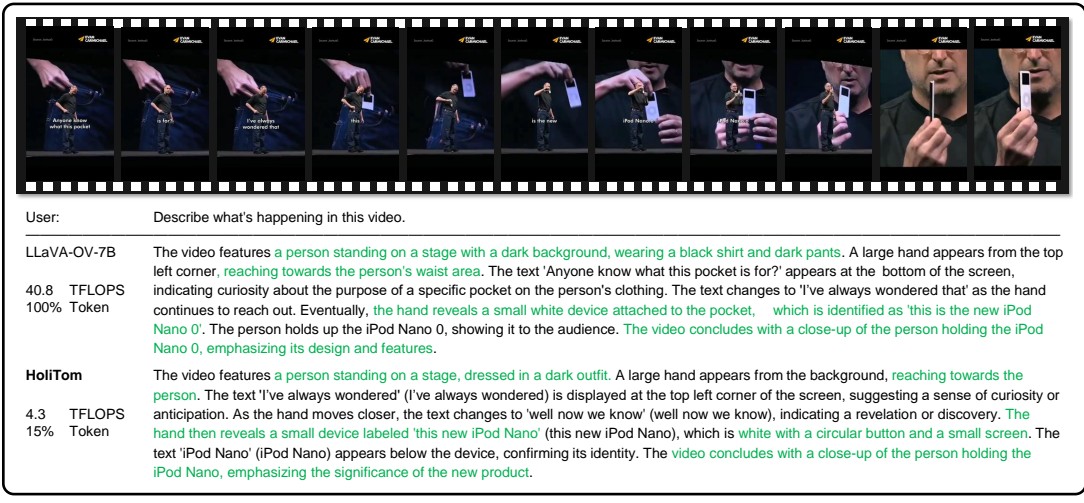

Figure 11: **Qualitative generation comparison.** Green indicates correctly detailed descriptions. Our method achieves high-quality, accurate text generation even when retaining only 15% of input tokens.

