# OpenReview forum: "HoliTom: Holistic Token Merging for Fast Video Large Language Models"
_NeurIPS.cc/2025/Conference — NeurIPS 2025 poster_

### Official Review · Reviewer_uHRM · 2025-06-21

**Clarity:** 4
**Significance:** 3
**Originality:** 3
**Rating:** 5
**Confidence:** 4

**Summary:**

This paper proposes HoliTom, a training-free token-compression framework that improves the inference-time efficiency of video LLMs. HoliTom combines outer-LLM reduction—which performs global optimization–based temporal merging and attention/cluster-based spatial merging—with inner-LLM reduction, which merges visual tokens at every layer using a fixed retention ratio instead of simply dropping redundant tokens. The method attains state-of-the-art results while requiring fewer FLOPs than prior token-compression approaches.

**Questions:**

Apart from three questions in weakness part, I have additional following questions:

4. For inner-LLM compression, the method uses the attention score of the “last token.” Does this refer to the last prompt token (visual + text) or the last visual token only? Have the authors compared the two variants, and if so, what did you observe?
5. Suppose we sample more frames yet keep overall FLOPs constant (eg. 100%) after applying HoliTom. Would this yield better performance than using fewer frames with the vanilla model?

If the authors address the concerns above, I believe the paper will be further solidified, and I will raise my score.

**Ethical Concerns:**

["NO or VERY MINOR ethics concerns only"]

**Final Justification:**

Overall, this paper proposes three key designs to achieve stronger training-free token compression for video VLMs.

In my view, the novelty gap between this work and prior or concurrent papers is not marginal. Focusing on a training-free token-compression paradigm, the paper introduces three key innovations that together deliver state-of-the-art performance and efficiency. Given the limited toolbox for further improving token compression and the intense competition in this area, these advances are both meaningful and valuable.

The rebuttal also provides convincing evidence of the method’s generalization capability and supplies the necessary technical details. Accordingly, I have kept my rating and recommend acceptance.

**Limitations:**

- Multi-round conversations. Can HoliTom handle multi-turn dialogue in video LLMs? If a follow-up query arrives, does it affect the validity of inner-LLM merging?
- Extremely long videos. Have the authors tested HoliTom on very long inputs (e.g., Qwen 2.5-VL with 768 frames)? What efficiency and accuracy gains does HoliTom provide compared with no reduction in that setting?

**Paper Formatting Concerns:**

No concern.

**Quality:**

3

**Strengths And Weaknesses:**

**Strengths**
1. **Effectiveness and efficiency.** Both outer-LLM and inner-LLM merging clearly improve performance, as evidenced by the ablations in Table 4. The temporal-merging objective is novel, and solving it via dynamic programming is elegant and effective.
2. **Comprehensive analysis.** Section 4 offers detailed experiments that thoroughly validate the proposed method. The inference-efficiency study highlights HoliTom’s advantage over previous techniques.
3. **Clarity of writing.** The paper is well written and easy to follow.

**Weaknesses**
1. **Missing spatial-merging ablation.** For non-redundant temporal tokens, the paper uses an attention-based strategy; for redundant tokens, it adopts DPC-KNN clustering. An ablation justifying these design choices is needed.
2. **Lack of qualitative visualizations.** Visual examples showing token reduction at each stage would help readers better understand how each component works.
3. **Equation (5) needs more explanation.** The derivation and intuition behind this equation should be clarified.

Minor Issues
- L156: “an” → “a”
- L198: “spatial-temporal” → “spatio-temporal”

---

> ### Author Rebuttal · Authors · 2025-07-31
>
> Thank you so much for the detailed and constructive comments. We address your concerns as follows.
>
> ---
>
> >**W1**: Missing spatial-merging ablation. For non-redundant temporal tokens, the paper uses an attention-based strategy; for redundant tokens, it adopts DPC-KNN clustering. An ablation justifying these design choices is needed.
>
> **A1**: We thank the reviewer for this valuable suggestion. We agree that an ablation study is essential to justify our design choices for the spatial-merging strategy. Following this advice, we have conducted the experiment, and the results are presented below.
>
> Here, Attn refers to using only the attention-based strategy for all tokens, while DPC-KNN uses only the clustering algorithm. Mixed is our proposed hybrid strategy.
>
> | Method  | MVBench | EgoSchema | LVBench | VideoMME | Avg. Score |
> |---------|---------|-----------|---------|----------|------------|
> | Attn    |**58.4**| **60.9**   | 55.9    | 57.4     | 58.1       |
> | DPC-KNN | 57.4    | 59.6      | 53.9    | 56.6     | 56.9       |
> | Mixed   |**58.4**| **60.9** | **56.2**  | **58.3** | **58.5**   |
>
> The results clearly show that our mixed strategy achieves the best performance. Our design principle is as follows:
>
> For non-redundant tokens, the encoder's self-attention scores serve as an **excellent prior** for visual importance within a single frame. A compression strategy guided by these intra-frame scores is therefore highly effective.
>
> However, redundant tokens are different, as they are formed by merging tokens from adjacent frames. Since the self-attention mechanism operates in encoder (siglip) only within a single frame, using these attention scores as a metric for the new, merged tokens lacks a solid theoretical basis. For this reason, we adopted DPC-KNN clustering to group these merged tokens based on their feature similarity, which is a more principled approach for this context. As noted in the paper (line 163), this design consideration is central to our method.
>
> We sincerely appreciate the reviewer's feedback. This supplementary experiment has allowed us to substantiate the rationale for our method more rigorously and has significantly strengthened our paper.
>
> ---
>
> >**W2**: Lack of qualitative visualizations. Visual examples showing token reduction at each stage would help readers better understand how each component works.
>
> **A2**: We completely agree that visualizations would significantly enhance the reader's understanding of our proposed token reduction process. We were unable to include figures due to the NeurIPS'25 rebuttal guidelines, but we are willing to incorporate these key visualizations in the next version. We appreciate you pointing out this opportunity for improvement.
>
> ---
>
> >**W3**: Equation (5) needs more explanation. The derivation and intuition behind this equation should be clarified.
>
> **A3**: This method for selecting cluster centers is based on the well-established Density Peak Clustering (DPC) algorithm, originally proposed by Rodriguez and Laio in Science (2014). The fundamental idea is that cluster centers are characterized by two key properties:
>
> - They are surrounded by neighbors with lower local density (i.e., they are local density maxima).
>
> - They are at a relatively large distance from any other points with a higher local density.
>
> The equations you referenced are designed to mathematically identify points that satisfy these two conditions.
>
> Intuition Breakdown
> Local Density $ρ_i$: This term measures how dense the feature space is around a given token $v_i$. A high value of $ρ_i$ indicates that the token is in a crowded region, surrounded by many other similar tokens. This suggests it's part of a potential cluster.
>
> Separation Distance $δ_i$: This term measures the minimum distance from token $v_i$ to any other token with a higher local density. A large $δ_i$ is significant: it means that even though $v_i$ is in a dense area, it is also the densest point in its local neighborhood, making it a "peak" in the density landscape. For the globally densest point, $δ_i$ is its maximum distance to any other token.
>
> Final Score $γ_i=ρ_i×δ_i$: The final score $γ_i$ effectively combines these two concepts. To be chosen as a cluster center, a token must not only be in a dense region (high $ρ_i$) but also be a clear density peak (high $δ_i$). By multiplying them, we find tokens that are simultaneously local density maxima and are well-separated from other, even denser points. These are the ideal candidates for cluster centers.
>
> In summary, this method provides a robust and non-parametric way to automatically find cluster centers by identifying the peaks in the data's density distribution. We will add this detailed explanation and the proper citation to the manuscript to eliminate any confusion.
>
> ---
>
> >**Q1**: For inner-LLM compression, the method uses the attention score of the “last token.” Does this refer to the last prompt token (visual + text) or the last visual token only? Have the authors compared the two variants, and if so, what did you observe?
>
> **A1**: The "last token" we utilize for attention-based compression is indeed the final token of the full input sequence, comprising both visual and textual elements.
>
> Our reason for this choice is that the final token of the complete sequence is positioned **closest** to the tokens that are to be generated. We believe its hidden state therefore encapsulates a more comprehensive understanding of the entire query (both visual and textual context), making it the most suitable anchor for calculating attention scores for compression. While we have not performed a direct empirical comparison, we appreciate you bringing this to our attention, as it is a valuable consideration for future analysis.
>
> ---
>
> >**Q2**: Suppose we sample more frames yet keep overall FLOPs constant (eg. 100%) after applying HoliTom. Would this yield better performance than using fewer frames with the vanilla model?
>
> | Method  | Frames | Prefilling FLOPs (T)  | FLOPs Ratio  | Avg Score |
> |---------|--------|-----------------------|--------------|-----------|
> | ov-7b   | 32     | 40.8                  | 100%         | 58.4      |
> | Holitom | 32     | 5.2                   | 12.7%        | 58.5      |
> | Holitom | 64     | 10.8                  | 26.5%        | 58.9      |
> | Holitom | 128    | 23.1                  | 56.6%        | 59.3      |
>
> **A2**: **Yes**, sampling more frames with HoliTom, while keeping the total FLOPs constant or even less, yields significantly better performance than a vanilla model operating on fewer frames.
>
> We also note that as the number of input frames grows, the computational overhead from the vision encoder becomes non-negligible. This trade-off can be seen in Figure 5 of our paper.
>
> ---
>
> >**L1**: Multi-round conversations. Can HoliTom handle multi-turn dialogue in video LLMs? If a follow-up query arrives, does it affect the validity of inner-LLM merging?
>
> **A1**: We appreciate you pointing this out. The validity of the merged tokens in a multi-turn scenario is indeed a key consideration that we carefully addressed in our work. The core of the issue lies in the distinction between our two token compression strategies:
>
> - **Inner-LLM Merging**: As you rightly suggest, this approach can be affected by follow-up queries. Because it computes joint attention over both visual and text tokens, the resulting compressed representation is influenced by the initial query. A new turn in the dialogue would ideally require re-computation.
> - **Outer-LLM Merging**: In contrast, this method is designed to be question-agnostic. The compression is performed independently of the language input, making it robust and consistently valid across multiple conversational turns.
>
> **Recognizing the distinct implications** of these two methods for multi-turn dialogue, we report the results for HoliTom (which uses inner-LLM merging) and HoliTom(w/o M) **separately** in our paper. Our intention is to provide the research community — especially those focused on downstream multi-turn Q&A tasks — with better understanding.
>
> >**L2**: Extremely long videos. Have the authors tested HoliTom on very long inputs (e.g., Qwen 2.5-VL with 768 frames)? What efficiency and accuracy gains does HoliTom provide compared with no reduction in that setting?
>
> | Method     | Token | Ratio | FLOPs  | FLOPs (%) |       |        |      | VideoMME | Avg. |
> |------------|-------|-------|--------|-----------|-------|--------|------|----------|------|
> |            |       | %     | TFLOPs | %         | Short | Medium | Long | Overall  | %    |
> | Qwen2.5-VL | 18442 | 100   | 377.2  | 100  | 77.4  | 68.1   | 55.6 | 67.0     | 100  |
> | +Visionzip | 9221  | 50.0  | 154.5  | 41.0 | **74.9**  | **66.6**   | 55.7 | 65.7     | 98.1 |
> | +PruneVid  | 9173  | 49.7  | 153.5  | 40.7 | 72.3  | 64.8   | 54.7 | 63.9     | 95.4 |
> | +HoliTom   | **6513**  | **34.9**  | **102.0**  | **27.0** | 74.4  | 66.4   | **56.4** | **65.8**     | **98.2** |
> | +Visionzip | 4610  | 25.0  | 68.7   | 18.2 | **73.1**  | 63.3   | 55.9 | 64.1     | 95.7 |
> | +PruneVid  | 4632  | 25.1  | 69.1   | 18.3 | 69.3  | 61.1   | 53.2 | 61.2     | 91.3 |
> | +HoliTom   | **4504**  | **24.4**  | **66.9**   | **17.7** | 72.7  | **65.7**   | **56.1** | **64.8**     | **96.7** |
>
> **A2**: We tested HoliTom on very long inputs (768 frames) on the comprehensive VideoMME benchmark, which includes a dedicated sub-evaluation set for long videos. This allows for a direct assessment of performance on extended sequences.
>
> Our findings on this "Long" video subset confirm the effectiveness of our approach. Specifically, HoliTom achieves a 0.5% performance increase over the baseline, even while retaining only 25% of the original tokens.
>
> ---
>
> *Thank you again for helping us improve our paper! We are **actively available** during the next author-reviewer discussion period. Should you have any further questions, please let us know!*

---

> > ### Author Response · Authors · 2025-08-04
> > **Kindly requesting feedback**
> >
> > Dear Reviewers,
> >
> > Thank you once again for your valuable comments on our submission. We have posted responses to the proposed concerns and included additional analysis and experiments.
> >
> > We totally understand that this is quite a busy period, so we deeply appreciate it if you could take some time to return further feedback on whether our response solves your concerns. If there are any other comments, we will try our best to address them.
> >
> > Best regards,
> >
> > The Authors.

---

> > > ### Comment · Reviewer_uHRM · 2025-08-04
> > >
> > > Thanks for the detailed response. It clearly addresses all of my concerns.
> > >
> > > I have also considered the other reviewers’ comments. In my view, the novelty gap between this work and prior or concurrent papers is not marginal. Focusing on a training-free token-compression paradigm, the paper introduces three key innovations that together deliver state-of-the-art performance and efficiency. Given the limited toolbox for further improving token compression and the intense competition in this area, these advances are both meaningful and valuable.
> > >
> > > The rebuttal also provides convincing evidence of the method’s generalization capability and supplies the necessary technical details. Accordingly, I have kept my rating and recommend acceptance.

---

### Official Review · Reviewer_jPEX · 2025-06-24

**Clarity:** 3
**Significance:** 2
**Originality:** 2
**Rating:** 4
**Confidence:** 4

**Summary:**

This paper introduces HoliTom, a holistic token merging method for video LLMs that integrates spatial, temporal, and inner-LLM token merging. HoLiTom employs: (1) Global redundancy-aware temporal merging, which reduces temporal redundancy by merging tokens into their first occurrence; (2) Spatial merging via clustering tokens based on global feature representations; and (3) Inner-LLM merging, where redundant tokens are averaged rather than discarded. Evaluated on multiple video understanding benchmarks, HoLiTom achieves high efficiency and strong performance.

**Questions:**

* Please refer to the Weaknesses section.
* If have time, an interesting extension would be evaluating HoliTom on training-free video LLMs (e.g., image-trained models like LLaVA-1.6 applied to videos, e.g. SF-LLaVA [1] or TS-LLaVA). Since these models lack explicit temporal knowledge from video training, does the token merging strategy still prove effective?


[1] Mingze Xu, Mingfei Gao, Zhe Gan, Hong-You Chen, Zhengfeng Lai, Haiming Gang, Kai Kang, Afshin Dehghan. (2024) SlowFast-LLaVA: A Strong Training-Free Baseline for Video Large Language Models.

[2] Tingyu Qu, Mingxiao Li, Tinne Tuytelaars, Marie-Francine Moens. (2024) TS-LLaVA: Constructing Visual Tokens through Thumbnail-and-Sampling for Training-Free Video Large Language Models

**Ethical Concerns:**

["NO or VERY MINOR ethics concerns only"]

**Final Justification:**

The rebuttals have addressed most of my concerns, and I believe the paper makes a valuable contribution to the subfield of video understanding. Having read the rebuttals and other reviews, I remain somewhat on the borderline, but I support the paper’s acceptance. If accepted, it would be a welcome addition to the field.

**Limitations:**

yes

**Quality:**

3

**Strengths And Weaknesses:**

**Strengths**
* The paper is clearly-written and well-organised. The method is quite intuitive.
* The method considers multiple aspects while improving efficiency, including spatial, temporal and inner-LLM redundancy. Overall, the method offers a comprehensive solution to token redundancy in video understanding tasks.
* Performance wise, strong performance and high efficiency are obtained as compared to previous methods when evaluating with video LLMs.

**Weaknesses**
* If I understand correctly, the core idea is to merge/average out redundant information based mainly on the global representations of the visual tokens (either the similarity calculation in temporal merging, CLS replacement in spatial merging or the inner-LLM merging). This approach seems effective for redundancy reduction and is highly favourable in terms of efficiency, but could it risk losing fine details (e.g., small objects) or become overly aggressive for longer videos?
* Following up on the longer videos, I noticed that on LongVideoBench HoliTom either matches or underperforms compared to prior methods, unlike its superior results on shorter videos (e.g., MVBench). Could you elaborate on this? Is this a limitation inherent to the method’s design, or are there other factors at play?
* I’m curious about HoliTom’s performance on tasks requiring fine-grained object-level reasoning. Given the method’s aggressive merging strategy, could you discuss whether it preserves sufficient detail for such scenarios? Additional analyses would be helpful.

---

> ### Author Rebuttal · Authors · 2025-07-31
>
> Thank you so much for the detailed and constructive comments. We address your concerns as follows.
>
> ---
>
> > **W1**: If I understand correctly, the core idea is to merge/average out redundant information based mainly on the global representations of the visual tokens (either the similarity calculation in temporal merging, CLS replacement in spatial merging or the inner-LLM merging). This approach seems effective for redundancy reduction and is highly favourable in terms of efficiency, but could it risk losing fine details (e.g., small objects) or become overly aggressive for longer videos?
>
> > **W3**: I’m curious about HoliTom’s performance on tasks requiring fine-grained object-level reasoning. Given the method’s aggressive merging strategy, could you discuss whether it preserves sufficient detail for such scenarios? Additional analyses would be helpful.
>
> **A1,3**: (The above two weaknesses are similar. We respond to them collectively here.) Thank you for your interest and accurate summary of our method. We appreciate the opportunity to elaborate on this important point.
>
> **Details Protection**
>
> Your concern about the potential loss of fine-grained details is indeed a critical consideration for any token compression strategy. However, HoliTom not only preserves these details but can actually enhance the model's ability to perceive them.
>
> To substantiate this, we would like to draw your attention to our performance on the **3 object-level subtasks** (OE, OI, OS) in MVBench. With compression rates of 10-25%, HoliTom leads to a **notable increase** in performance. This outcome would be unlikely if our method are discarding critical information about small objects. Instead, it serves as strong evidence of our method's ability to retain fine-grained details effectively.
>
> > OE: Object Existence - Example: Are there any moving green objects when the video ends? (A) not sure (B) yes (C) no
>
> > OI: Object Interaction - Example: Which object was tidied up by the person? (A) broom (B) cabinet (C) blanket (D) table
>
> > OS: Object Shuffle - Example: Where is the hidden object at the end of the game from the person's point of view? (A) Under the first object from the left. (B) Under the third object from the left. (C)Under the second object from the left.
>
> |             | FLOPs Ratio | OE   | OI   | OS   | Overall |
> |-------------|-------------|------|------|------|---------|
> | LLaVA-OV-7B | 100%        | 57.5 | 84.0 | 35.5 | 58.3    |
> | +HoliTom    | 17.4%       | 61.0 | 83.5 | 36.5 | 58.4    |
> | +HoliTom    | 14.2%     |**63.0**| 84.0 | 37.5| **58.7** |
> | +HoliTom    | 10.5%      | 60.5 |**84.5**|**38.0**| 58.1 |
> | +HoliTom    |**6.9%**     | 60.0 | 81.0 | 37.0 | 57.3    |
>
> This counterintuitive performance boost is discussed in our paper (line 264):
> > **The value of key information over exhaustive information.** As we state: Excessive, irrelevant, or redundant data acts as noise, obscuring essential signals critical for effective processing. This information overload impedes the capacity of the model to accurately identify and process critical details, thereby degrading understanding and response generation.
>
> Frankly, I concede that extreme token compression will inevitably degrade performance across the board, including detail-oriented understanding. We cannot expect a handful of tokens to carry the informational capacity of a massive dataset.
>
> **Concerns about the method's ability for longer video understanding**
>
> To test Holitom's ability on long video understanding, we followed the recommendations from the Qwen2.5-VL technical report, employing dynamic resolution and increasing the maximum input to 768 frames. We capped the video token count at 24,576, as suggested by the report. This inherently imposes a stricter representational constraint of 32 tokens per frame (24,576 tokens / 768 frames), a significant reduction compared to LLaVA-Video's 169 tokens per frame.
>
> Even under these stringent conditions, HoliTom demonstrated **remarkable** performance. Notably, for the long-video sub-task of VideoMME, HoliTom **surpassed** the original baseline's performance under both compression settings, underscoring its potential for long-video processing. In this latter case, the average number of tokens per frame was compressed to 8.
>
> | Method     | Token |       | FLOPs  |           |       |        |      | VideoMME | Avg. |
> |------------|-------|-------|--------|-----------|-------|--------|------|----------|------|
> |            |       | %     | TFLOPs | %         | Short | Medium | Long | Overall  | %    |
> | Qwen2.5-VL | 18442 | 100   | 377.2  | 100       | 77.4  | 68.1   | 55.6 | 67.0     | 100  |
> | +Visionzip | 9221  | 50.0  | 154.5  | 41.0      |**74.9**|**66.6**| 55.7 | 65.7     | 98.1 |
> | +PruneVid  | 9173  | 49.7  | 153.5  | 40.7      | 72.3  | 64.8   | 54.7 | 63.9     | 95.4 |
> | +HoliTom   |**6513**|**34.9**|**102.0**|**27.0**| 74.4  | 66.4   |**56.4**| **65.8** | **98.2** |
> | +Visionzip | 4610  | 25.0  | 68.7   | 18.2      |**73.1**| 63.3  | 55.9 | 64.1     | 95.7 |
> | +PruneVid  | 4632  | 25.1  | 69.1   | 18.3      | 69.3  | 61.1   | 53.2 | 61.2     | 91.3 |
> | +HoliTom   |**4504**|**24.4**|**66.9**|**17.7** | 72.7  |**65.7**|**56.1**| **64.8**| **96.7**
>
> ---
>
> > **W2**: Following up on the longer videos, I noticed that on LongVideoBench HoliTom either matches or underperforms compared to prior methods, unlike its superior results on shorter videos (e.g., MVBench). Could you elaborate on this? Is this a limitation inherent to the method’s design, or are there other factors at play?
>
> | Model          | OV-7B | VisionZip | PruneVid | FastVID | HoliTom | OV-72B | VisionZip | PruneVid | HoliTom | Vid-7B | VisionZip | HoliTom |
> |----------------|-------------|------------|-----------|----------|----------|--------------|------------|-----------|----------|--------------|------------|----------|
> | MVBench        | 58.3        | 56.5       | 56.8      | 56.0     | **58.1** | 60.9         | 58.4       | 56.8      | **58.7** | 60.4         | 56.7       | **57.7** |
> | LongVideoBench | 56.4        | 54.4       | 55.4      | 56.2     | **56.4** | 62.7         | **57.4**   | **57.4**  | 57.2     | 58.9         | 54.7       | **56.2** |
>
> **A2**: Thanks for the question, but there might be a misunderstanding here.
>
> As shown in the paper (Table 2,3 ), on 3 different LLM backbones and at multiple ratios, HoliTom *surpasses* the previous SoTA method in both MVBench and LongVideoBench benchmarks in the majority when tested. This result underscores its exceptional ability to maintain high-fidelity comprehension for videos of all lengths.
> ﻿
> ---
>
> > **Q1**: If have time, an interesting extension would be evaluating HoliTom on training-free video LLMs (e.g., image-trained models like LLaVA-1.6 applied to videos, e.g., SF-LLaVA or TS-LLaVA). Since these models lack explicit temporal knowledge from video training, does the token merging strategy still prove effective?
>
> **Q1-Ans**: Thank you for your insightful suggestion! We believe that our method, HoliTom, would indeed remain effective for training-free video LLMs.
>
> We find this to be a fascinating direction for empirical validation. Due to device constraints, we are unable to provide experimental results for this specific setting within the rebuttal period. We give our analysis below:
>
> The effectiveness of our approach is rooted in principles that are not dependent on the model having explicit temporal knowledge from video-specific training. We can break this down based on our two-stage compression:
>
> - **Outer-LLM Token Compression**: This stage targets the inherent redundancy within the visual data itself. For instance, in a model like LLaVA-OneVision that uses a vision encoder like SigLIP, video frames are typically treated as a batch of images (i.e., the `batch_size` dimension in a tensor like `[batch_size, head_num, seq_len, head_dim]`). The frames are encoded without explicit cross-frame computation, meaning the model lacks inherent temporal knowledge at this stage. Our outer-LLM compression operates on these encoded representations after they have been processed by the vision encoder. The redundancy it compresses is therefore the semantic and visual overlap between frames, which is a fundamental property of the video data itself, not a feature learned by the model.
> - **Inner-LLM Token Compression**: This stage leverages the **sparse attention** phenomenon common in large vision-language models. We suggest that this sparsity primarily arises from two factors: the misalignment between visual and text, and the redundant visual representations. These factors are characteristic of the multi-modal data and architecture themselves, rather than being dependent on whether the model was trained on video datasets. Therefore, since the underlying causes of sparse attention are not unique to video-trained models, we expect the inner-LLM compression to be effective in identifying and merging non-essential tokens in a training-free context as well.
>
> ---
>
> *Thank you again for helping us improve our paper! We are **actively available** during the next author-reviewer discussion period. Should you have any further questions, please let us know!*

---

> > ### Author Response · Authors · 2025-08-04
> > **Kindly requesting feedback**
> >
> > Dear Reviewers,
> >
> > Thank you once again for your valuable comments on our submission. We have posted responses to the proposed concerns and included additional analysis and experiments.
> >
> > We totally understand that this is quite a busy period, so we deeply appreciate it if you could take some time to return further feedback on whether our response solves your concerns. If there are any other comments, we will try our best to address them.
> >
> > Best regards,
> >
> > The Authors.

---

> > ### Comment · Reviewer_jPEX · 2025-08-05
> >
> > Thanks for the detailed response. In general, I'm satisfied with the response. However, for W2, especially in Table 2, I do not see a clear advantage of HoliTom over FastVID on LongVideoBench under different FLOPs ratios. Since in Table 3, no results on FastVID are reported, and in Table 2, no results of FastVID on EgoSchema are reported, I think it's still difficult to claim that  HoliTom outperforms previous methods for long video understanding.

---

> > > ### Author Response · Authors · 2025-08-06
> > > **Thank you! Further Responses**
> > >
> > > Thank you for agreeing with our previous response. To thoroughly address your concern, we have urgently conducted additional experiments by reproducing FastVID's results on EgoSchema.
> > >
> > > | Method  | FLOPs (%) | EgoSchema |
> > > |---------|-----------|-----------|
> > > | FastVID | 21.3      |  59.4    |
> > > | HoliTom | **17.4**      | **61.2**      |
> > > | FastVID | 17.2      |  59.3   |
> > > | HoliTom | **14.2**      | **61.0**      |
> > > | FastVID | 12.7      |  58.8    |
> > > | HoliTom | **10.5**      |  **61.2**     |
> > > | FastVID | 8.3       | 58.5      |
> > > | HoliTom | **6.9**       | **61.2**      |
> > >
> > >
> > > As presented in the table above, our results show that HoliTom consistently outperforms FastVID across all four compression ratios on the EgoSchema benchmark (ov-7b).
> > >
> > > EgoSchema is a closed-source benchmark, meaning results are obtained only by uploading a `result.json` file for official evaluation. Thus, we attribute this advantage to HoliTom's superior ability to preserve critical information during long video token compression.
> > >
> > > Furthermore, we believe the following points provide crucial context:
> > >
> > > - *Concurrent Work*: Despite being considered concurrent work—with its v1 uploaded to arXiv on March 14th, less than three months before the NeurIPS '25 submission deadline (May 15th)—we still reported FastVID's results in our publication out of respect for this recent research.
> > >
> > > - *Limited Reproducibility*: We are unable to reproduce all of FastVID's results for a complete comparison in Table 3. Its authors only open-sourced the code for their ov-7B model in early June, while the code for the ov-72B and vid-7B models remains unavailable. Consequently, we have focused on supplementing Table 2 with the directly comparable EgoSchema results.
> > >
> > > - *Advantage on Other Datasets*: As shown in our original manuscript, HoliTom achieves superior results compared to FastVID on the MVBench and VideoMME benchmarks in most cases.
> > >
> > > - *Revisiting the LongVideoBench Results*: The original score on LongVideoBench for ov-7b is 56.4. Notably, HoliTom, even after compressing video tokens to just 10%, still achieves a score of 56.3. This suggests that the 10% compression ratio on LongVideoBench is not a sufficiently challenging task for HoliTom and may therefore mask its advantage over FastVID.
> > >
> > > Thank you again for your valuable feedback. We are committed to solving all of your concerns.

---

> > > > ### Comment · Reviewer_jPEX · 2025-08-06
> > > >
> > > > Thank you for providing the additional results, which further strengthen the claims made in the paper. I am satisfied with the rebuttal overall and support the paper’s acceptance.

---

### Official Review · Reviewer_N3Be · 2025-06-30

**Clarity:** 3
**Significance:** 2
**Originality:** 2
**Rating:** 4
**Confidence:** 4

**Summary:**

In this work, the authors propose a training-free Holistic Token Merging method to boost computation efficiency for video MLLMs. Basically, it consists of three key parts, including Temporal Merging, Spatial Merging, and Inner-LLM Merging.

**Questions:**

Please see the weakness section in details.

1 Limited Novelty: Please clarify the key contributions in the proposed merging method, compared to the existing methods, e.g., VideoChat-Flash, Video-XL, MovieChat, etc.

2 Insufficient Comparison: To further show its effectiveness, it is necessary to show compasion with the sota methods metioned above.

3 More Investigation: Video QA is only one task in long video understanding. It is interesting to show if this method works for other long video understanding tasks like temporal grounding or  Needle-in-A-Video Haystack.

**Ethical Concerns:**

["NO or VERY MINOR ethics concerns only"]

**Final Justification:**

Thanks for the rebuttal. The authors have addressed most of my main concerns. I change my rating as borderline accept.

**Limitations:**

Please see the weakness section.

**Quality:**

3

**Strengths And Weaknesses:**

*Strengths

1 The efficiency of video MLLMs is an important research problem, especially for long video understanding. The authors attempt to address this issue via token merging.

2 The paper is well organized with good structure.

*Weaknesses

1 The novelty is actually limited. The merging strategies ( including temporal, spatial and inner-LLM) in this work are straightforward, according to similarity comparsion. They are not foundamentally different from those merging strategies of the existing methods e.g., VideoChat-Flash, Video-XL, MovieChat, etc. Please clarify the key contributions in the proposed merging method.

2 Moreover, in the experiment section, comparison with these sota methods (e.g., VideoChat-Flash, Video-XL, MovieChat, etc) is missing.  To further show its effectiveness, it is necessary to show such compasion.

3 Besides of video QA, does this method work for other long video understanding tasks like temporal grounding (e.g., Charades-STA)  or  Needle-in-A-Video Haystack tasks in VideoChat-Flash.

---

> ### Author Rebuttal · Authors · 2025-07-31
>
> Thank you so much for the detailed and constructive comments. We address your concerns as follows.
>
> ---
>
> > **Q1**: The novelty is limited. The merging strategies (including temporal, spatial, and inner-LLM) in this work are straightforward, according to similarity comparison. They are not fundamentally different from those merging strategies of the existing methods, e.g., VideoChat-Flash, Video-XL, MovieChat, etc. Please clarify the key contributions in the proposed merging method.
>
> **A1**: We understand the concern that the merging scheme used by other works (VideoChat-Flash, Video-XL, MovieChat) may make our work "appear" similar to them. However, our work is actually distinct from these works. Before detailing our novel contributions, it is worth noting the key differences that separate HoliTom from the methods mentioned above:
>
> - VideoChat-Flash, Video-XL, and MovieChat are primarily **training-based** methods. Their performance is a result of the entire model architecture, the training data, and the specific training strategies employed, as well as their token compression techniques. The associated **training costs are non-trivial**.
>
> - In contrast, HoliTom is proposed not as a new standalone model, but as a **training-free**, **parameter-free**, and **plug-and-play** token compression framework. Its purpose is to be seamlessly inserted into existing, powerful video LLMs to help them achieve a superior balance between performance and efficiency.
>
> In light of this crucial distinction, direct comparison between our training-free method vs. their training-based methods might not fully capture our contribution.
>
> With this clarification, we would like to address your primary concern by detailing the key contributions and novelty within our proposed methodology:
>
> **Contributions**:
>
> - **Optimal Temporal Merging via Dynamic Programming**: We introduce a more principled approach to segmenting and compressing visual tokens. Existing methods rely on naive fixed-cliping or clustering on abstracted frame tokens (a pooled or [CLS] token), which ignore the rich, underlying visual tokens. Our key innovation is to solve this by formulating it as a global optimization problem and solving it with Dynamic Programming (DP). By operating directly on the complete set of fine-grained, raw visual tokens, our DP algorithm finds the optimal solution that simultaneously achieves the dual objectives of maximum temporal redundancy pruning and the most coherent semantic video segmentation.
> - **Synergistic Inner-LLM Merging that Preserves Information**: Common techniques like token dropping, while simple, risk irreversible information loss. Synergy with our outer-LLM merging and considering information loss, we merge related tokens in the LLM’s deeper layers. This condenses rich details into representative tokens and mitigates the performance degradation caused by dropping.
> - **SoTA Efficiency-Performance Trade-off**: Extensive evaluations on LLaVA-OneVision and LLaVA-Video demonstrate that our integrated pruning framework achieves a state-of-the-art efficiency-performance trade-off, significantly reducing computational costs and accelerating real-world inference while preserving model performance.
>
> ---
>
> > **Q2**: Insufficient Comparison: To further show its effectiveness, it is necessary to show a comparison with the SOTA methods mentioned above.
>
> **A2**: Per the suggestion, we add comparisons on Qwen2.5-VL. Results are below.
>
> | Method               | T/F     | Res.   | Token ratio (%) | VideoMME | Charades-STA mIoU |
> |----------------------|---------|--------|-----------------|----------|-------------------|
> | MovieChat            |         |        | -               | 38.2     | <10               |
> | Video-XL             |         |        | -               | 55.5     | -                 |
> | VideoChat-Flash @224 | 16      | 50176  | -               | 64.0     | 48.4              |
> | VideoChat-Flash @448 | 16      | 200704 | -               | 65.3     | 48.0              |
> | Qwen2.5-VL+Visionzip | 16      | 25088  | 50.0            | 65.7     | 49.2              |
> | Qwen2.5-VL+PruneVid  | 16      | 25088  | 49.7            | 63.9     | 28.8              |
> | Qwen2.5-VL+HoliTom   | **~11** | 25088  | **34.9**        | **65.8** | **49.7**          |
> | Qwen2.5-VL+HoliTom   | 8       | 25088  | 24.4            |   64.8   | 45.9              |
>
> As seen, with less input image resolution, fewer tokens/frame, and potentially fewer input frames, Qwen2.5-VL+HoliTom **surpasses** VideoChat-Flash @448 on both the VideoMME and Charades-STA benchmarks.
>
> This result highlights HoliTom's strength as a plug-and-play module that enables powerful video LLMs to achieve a superior balance between performance and computational cost, *even being training-free*.
>
> ---
>
> > **Q3**: Besides video QA, does this method work for other long video understanding tasks like temporal grounding (e.g., Charades-STA) or Needle-in-A-Video Haystack tasks in VideoChat-Flash.
>
> **A3**:
> **Regarding the temporal grounding task** - Charades-STA, we are pleased to report that our approach surpasses the current SoTA training-free token compression methods at both evaluated compression ratios. We attribute this success to our temporal compress strategy, which, unlike methods such as PruneVid that disrupt temporal order, consciously preserves the temporal information of video frames, a critical aspect for this task.
>
> We do observe a performance decline at higher compression rates. The reasons are analyzed below.
>
> - **Analyses**: General video understanding (like QA) relies on interpreting what content is present in the video. Training-free compression methods benefit from the preliminary semantic alignment between visuals and text by the vision encoder. The attention scores in the encoder and clustering mechanisms can then effectively guide the condensation of this **existing semantic understanding**. In contrast, temporal grounding is about pinpointing when an event occurs. This requires the joint training of the vision encoder and LLM with the help of calibrated data. Neither vision encoder nor LLM can do this alone. This capability is not something that can be expected to emerge in a zero-shot manner; it **must be learned**.
>
> **Regarding Needle-in-a-Video-Haystack** tasks: We also appreciate the reviewer mentioning this challenging benchmark. At present, the backbone of our method, Qwen2.5-VL, is constrained to a maximum input of 768 frames. The significant computational overhead required for processing 1k-10k frames makes it infeasible for us to conduct these experiments on this task for the current submission. We are grateful for the suggestion and see this as an important future direction.
>
> ---
>
> *Thank you again for helping us improve our paper! We are **actively available** during the next author-reviewer discussion period. Should you have any further questions, please let us know!*

---

> > ### Comment · Reviewer_N3Be · 2025-08-04
> >
> > Thanks for the rebuttal. The authors have addressed most of my main concerns.

---

> ### Author Response · Authors · 2025-08-04
> **Kindly requesting feedback**
>
> Dear Reviewers,
>
> Thank you once again for your valuable comments on our submission. We have posted responses to the proposed concerns and included additional analysis and experiments.
>
> We totally understand that this is quite a busy period, so we deeply appreciate it if you could take some time to return further feedback on whether our response solves your concerns. If there are any other comments, we will try our best to address them.
>
> Best regards,
>
> The Authors.

---

> ### Author Response · Authors · 2025-08-04
> **Thank You for Your Feedback & Follow-up**
>
> Dear Reviewer,
>
> Thank you so much for your feedback on our rebuttal. We are glad to hear that we have addressed most of your main concerns.
>
> We are committed to thoroughly addressing your concerns. To ensure we haven't missed anything important, we would welcome the opportunity to clarify any *further* critical points you may have.
>
> Given that we have resolved the majority of your concerns, we would be very grateful if you would kindly consider *raising your score* in recognition of our rebuttal efforts and the resulting improvements.
>
> Thank you again for your time and consideration.
>
> Best regards,
>
> The Authors.

---

### Official Review · Reviewer_rsKW · 2025-07-03

**Clarity:** 3
**Significance:** 3
**Originality:** 2
**Rating:** 3
**Confidence:** 3

**Summary:**

The paper introduces HoliTom, a training-free token merging method to enhance the efficiency of video LLMs by compressing redundant visual tokens during video understanding tasks. The motivation stems from the high computational inefficiency of video LLMs due to the large number of tokens generated from numerous frames. HoliTom proposes a three-stage method: global redundancy-aware temporal merging to reduce temporal redundancy, spatial merging to compress spatial information, and inner-LLM merging to optimize token processing within the LLM. As a result, they reduce FLOPs to 6.9% while maintaining 99.1% of the baseline performance across various benchmarks, and ablation studies confirm the effectiveness of each module and segmentation strategy.

**Questions:**

See above

**Ethical Concerns:**

["NO or VERY MINOR ethics concerns only"]

**Limitations:**

See the weaknesses

**Quality:**

2

**Strengths And Weaknesses:**

Strengths:
1. The paper’s use of a combined temporal, spatial, and inner-LLM merging strategy. It is technically sound as it provides a comprehensive solution.
2. The paper achieves a 6.9% FLOPs reduction while retaining 99.1% performance, making it practical for real-world applications.
Weakness:
1. Instead of LLaVA series, applying the method to other video llm structure is necessary to showcase the generalization of the idea. Otherwise, I might suspect that it only works on simple (LLaVA) structures.
2. The paper lacks the explanation of why without inner-LLM could sometimes get the best performance.
3. The threshold for in line134 is manually crafted which is not a general method intrinsically.
4. While FLOPs and TTFT are measured, there’s no mention of actual inference time on different hardware setups. I’d like to see a comparison of real-time latency across GPUs to better assess practical deployment feasibility—can the authors provide this in their rebuttal?
5. The method handles 64 frames well, but it’s unclear how it performs with extremely long videos. I’m puzzled by this gap—could the authors test HoliTom on longer sequences to evaluate scalability and potential bottlenecks

---

> ### Author Rebuttal · Authors · 2025-07-31
>
> Thank you so much for the detailed and constructive comments. We address your concerns as follows.
>
> ---
>
> > **Q1**: "Instead of LLaVA series, applying the method to other video llm structure is necessary to showcase the generalization of the idea. Otherwise, I might suspect that it only works on simple (LLaVA) structures."
>
> **A1**: We select the frontier Qwen2.5-VL-7B as our base model, which notably supports dynamic resolutions and long video inputs of up to 768 frames. In a controlled comparison with other training-free token compression methods, we standardize the input to 128 frames and a fixed resolution, aligning the token count with that of LLaVA-Video (11648 tokens).
>
> | Method      | Token |      | FLOPs  |      |       |        |      | VideoMME | Avg. |
> |-------------|-------|------|--------|------|-------|--------|------|----------|------|
> |             |       | %    | TFLOPs | %    | Short | Medium | Long | Overall  | %    |
> | Qwen2.5-VL  | 11256 | 100  | 197.8  | 100  | 76.2  | 67.2   | 55.0 | 66.1     | 100  |
> | +Visionzip  | 2814  | 25.0 | 39.9   | 20.2 | 71.4  | 59.8   | 52.3 | 61.2     | 92.6 |
> | +PruneVid   | 2823  | 25.0 | 40.0   | 20.2 | 65.8  | 55.8   | 49.6 | 57.0     | 86.2 |
> | +HoliTom|**2800**|**24.9**|**39.7**|**20.1**|**72.0**|**61.3**|**52.9**|**62.1**|**93.9**|
>
> **Observation**: HoliTom achieves superior efficiency, retaining 93.9% of the original performance with only 20.1% of the original FLOPs. This demonstrates HoliTom's exceptional efficiency and its robust adaptability to new architectures. We believe this finding strongly supports the contributions of our work.
>
> ---
>
> > **Q2**: "The paper lacks the explanation of why without inner-LLM could sometimes get the best performance."
>
> **A2**: Without inner-LLM merging, a *greater* number of the original, less-compressed tokens are retained within the LLM. For a powerful LLM, it can be capable of understanding a larger number of tokens and can deliver better performance when provided with more of the original, uncompressed tokens.
>
> Besides, sometimes, retaining fewer tokens can also lead to better performance. Concurrent studies [*1] have also observed this phenomenon. We also provide an explanation for this phenomenon in line 264 of the paper, summarized as follows:
>
> > **The value of key information over exhaustive information.** As we state: Excessive, irrelevant, or redundant data acts as noise, obscuring essential signals critical for effective processing. This information overload impedes the capacity of the model to accurately identify and process critical details, thereby degrading understanding and response generation.
>
> - [*1] 2025, Arxiv, VideoChat-Flash: Hierarchical Compression for Long-Context Video Modeling
>
> ---
>
> > **Q3**: The threshold for in line 134 is manually crafted, which is not a general method intrinsically.
>
> **A3**: The threshold τ is grounded in a clear physical interpretation. It quantifies the similarity between tokens at the same spatial location in adjacent frames. A higher similarity score directly indicates that the feature representations of these frames are more alike, signifying temporal redundancy.
> ﻿
> Besides, we add sensitivity analysis experiments to further clarify this.  The table below shows that the results of the average performance on the four test datasets are *robust* to different choices of this hyperparameter -- As seen, at the setup of 15% token compression rate, the 5 different τ's achieve similar average performance (the std is only 0.2). This low sensitivity demonstrates that while τ is the only hyperparameter, it does not require meticulous tuning for each specific case, thus underscoring the generalizability of our method.
>
> | τ         | 0.5  | 0.6  | 0.7  | 0.8  | 0.9  | Standard deviation (std)  |
> |-----------|------|------|------|------|------|-----|
> | Ratio=15% | 58.1 | 57.9 | 58.1 | 58.5 | 58.2 | 0.2 |
>
> ---
>
> > **Q4**: "While FLOPs and TTFT are measured, there’s no mention of actual inference time on different hardware setups. I’d like to see a comparison of real-time latency across GPUs to better assess practical deployment feasibility—can the authors provide this in their rebuttal?"
>
> **A4**: The number of decoded tokens is variable. To provide a standardized and useful metric, our paper reports the average latency per decoded token. This allows for an estimation of the total inference time for any given output length using the formula:
>
>  Total Inference Time ≈ TTFT + (Average Latency per Token × Number of Decoded Tokens)
>
> To directly address your concern, we add an evaluation on the VideoMME benchmark to provide a comparison of total inference time. The table below presents the results for our method, HoliTom, versus the baseline on an A6000 GPU.
>
> | A6000       | TTFT    | sec/token | Latency | Avg. Score |
> |-------------|---------|-----------|---------|------------|
> | LLaVA-OV-7B | 1687 ms | 33.9 ms   | 191min  | 58.4       |
> | +Holitom  |**741 ms**|**25.8 ms** |**158min**| 57.9       |
>
> The results show that our method not only achieves a significant reduction in Time to First Token (TTFT) and latency per token but also reduces the total inference time for the entire benchmark by 33 minutes. This demonstrates a substantial performance advantage in a practical inference scenario, underscoring the real-world applicability and efficiency of our approach.
>
> It is also important to note that this evaluation was performed using the `lmms-eval` framework. Within this framework, a majority of the processing time is consumed by video decoding (pre-processing), which leads to low GPU utilization. Consequently, the latency improvements presented here are a conservative measure. We are confident that in more GPU-intensive applications, the performance gains from our method would be even more pronounced.
>
> We are committed to thoroughly validating our work, and we will be expanding our tests to include a wider range of hardware setups in future work.
>
> ---
>
> > **Q5**: The method handles 64 frames well, but it’s unclear how it performs with extremely long videos. I’m puzzled by this gap—could the authors test HoliTom on longer sequences to evaluate scalability and potential bottlenecks
>
> **A5**: Thanks for the insightful suggestion! We add the evaluations on Qwen2.5-VL.
>
> Specifically, we follow the recommendations from the Qwen2.5-VL technical report, employing dynamic resolution and increasing the maximum input to 768 frames. We cap the video token count at 24,576, as suggested by the report. This inherently imposes a stricter representational constraint of 32 tokens per frame (24,576 tokens / 768 frames), a significant reduction compared to LLaVA-Video's 169 tokens per frame.
>
> Results are below:
>
> | Method     | Token |       | FLOPs  |           |       |        |      | VideoMME | Avg. |
> |------------|-------|-------|--------|-----------|-------|--------|------|----------|------|
> |            |       | %     | TFLOPs | %         | Short | Medium | Long | Overall  | %    |
> | Qwen2.5-VL | 18442 | 100   | 377.2  | 100       | 77.4  | 68.1   | 55.6 | 67.0     | 100  |
> | +Visionzip | 9221  | 50.0  | 154.5  | 41.0      |**74.9**|**66.6**| 55.7 | 65.7     | 98.1 |
> | +PruneVid  | 9173  | 49.7  | 153.5  | 40.7      | 72.3  | 64.8   | 54.7 | 63.9     | 95.4 |
> | +HoliTom   |**6513**|**34.9**|**102.0**|**27.0**| 74.4  | 66.4   |**56.4**| **65.8** | **98.2** |
> | +Visionzip | 4610  | 25.0  | 68.7   | 18.2      |**73.1**| 63.3  | 55.9 | 64.1     | 95.7 |
> | +PruneVid  | 4632  | 25.1  | 69.1   | 18.3      | 69.3  | 61.1   | 53.2 | 61.2     | 91.3 |
> | +HoliTom   |**4504**|**24.4**|**66.9**|**17.7** | 72.7  |**65.7**|**56.1**| **64.8**| **96.7** |
>
>
> As seen, even under these stringent conditions, HoliTom demonstrates very promising performance. It maintains 98.2% of the original performance while using only 27% of the FLOPs. In an even more extreme scenario, it retains 96.7% of the performance with a mere 17.7% of the computational cost. Notably, for the long-video sub-task of VideoMME, HoliTom surpasses the original baseline's performance under both compression settings, underscoring its potential for long-video processing. In this latter case, the average number of tokens per frame is compressed to 8.
>
> ---
>
> *Thank you again for helping us improve our paper! We are **actively available** during the next author-reviewer discussion period. Should you have any further questions, please let us know!*

---

> ### Author Response · Authors · 2025-08-04
> **Kindly requesting feedback**
>
> Dear Reviewers,
>
> Thank you once again for your valuable comments on our submission. We have posted responses to the proposed concerns and included additional analysis and experiments.
>
> We totally understand that this is quite a busy period, so we deeply appreciate it if you could take some time to return further feedback on whether our response solves your concerns. If there are any other comments, we will try our best to address them.
>
> Best regards,
>
> The Authors.

---

> > ### Comment · Reviewer_rsKW · 2025-08-05
> >
> > Thanks for your rebuttal. Some explanations are sill unclear to me. Many details are missing in the paper and rebuttal. I will keep my original score.

---

> ### Author Response · Authors · 2025-08-05
> **Thank you! May you clarify the unclear points so that we can address them**
>
> Dear Reviewer rsKW,
>
> We greatly appreciate your time and effort in reviewing so far and helping us improve our work. We did our best to promptly address the concerns you raised within the scope of our previous response.
>
> We are sorry to see that you mentioned "*Some explanations are still unclear to me. Many details are missing in the paper and rebuttal*". Now we still have the time to address your concerns, so **could you please *specifically* let us know what the "some explanations" and "many details" you mentioned are**?
>
> Per the NeurIPS reviewer guidelines (https://neurips.cc/Conferences/2025/ReviewerGuidelines). Reviewers are supposed to `"Be specific. Do not make vague statements in your review, as they are unfairly difficult for authors to address"`.
>
> We understand you may be very busy at this point, and we truly appreciate your time. But the unclear comments cannot help us resolve your concerns, and cannot help us improve our work either. Thank you!
>
> Sincerely,
>
> Authors

---

### Comment · Area_Chair_UdfL · 2025-08-04

Dear Reviewers,

If you have not done so already, please review the authors' rebuttal to your comments and the other reviews.  Please submit any further questions for the authors promptly to allow time for discussion.

Please also remember to update your ratings and final justification if your concerns have been addressed. If ratings are not updated, clearly explain any remaining concerns in your final justification.

As a reminder, the author-reviewer discussion period ends on August 6th, 11:59 PM AoE.

Best, Your AC

---

### Note · Authors · 2025-08-12

We sincerely thank the ACs and reviewers for their time and constructive engagement, which has contributed to improving the clarity and overall quality of our work.

`Reviewer uHRM` **(Initial Rating:5, Confidence:4)** evaluated our work as *"elegant and effective"* and affirmed its novelty was *"not marginal"* and *"valuable"*, given the *"intense competition in this area"*. After the discussion, the reviewer noted that our response *"clearly addresses all of my concerns"* and consequently maintained a high rating and *"recommend acceptance"*.

`Reviewer jPEX` **(Initial Rating:4, Confidence:4)** found the method clear and efficient. In response to the request, the additional comparative results were provided, which the reviewer confirmed *"further strengthen the claims made in the paper"*. Stating he/she was *"satisfied with the rebuttal overall"*, the reviewer concluded *"support the paper’s acceptance"*.

`Reviewer N3Be` **(Initial Rating:3, Confidence:4)** prompted a discussion on our work's novelty compared to existing *trained models*. This allowed us to clarify the core distinction of our *training-free*, *plug-and-play* framework and, at their suggestion, provide new state-of-the-art comparisons. We are pleased that the reviewer confirmed we had *"addressed most of my main concerns"*. To ensure all issues were fully resolved, we explicitly invited any remaining critical points. As *no further points were raised*, we regard that we have fully addressed the reviewer's concerns.

`Reviewer rsKW` **(Initial Rating:3, Confidence:3)** acknowledged our method as *"technically sound"* and *"practical for real-world applications"*. The reviewer lastly stated that *"Some explanations are still unclear to me. Many details are missing in the paper and rebuttal."*. In our effort to fully address this, we requested specific details (*in line with NeurIPS reviewer guidelines*: reviewers should *"Be specific"*, because vague statements are *"unfairly difficult for authors to address"*). Finally, we did not receive the specific example, thus we could not answer the question.

Overall, the constructive dialogue *resolved all actionable concerns*. We thank all ACs and reviewers for their efforts.

---

### Decision · Program_Chairs · 2025-09-17

**Decision:**

Accept (poster)

**Comment:**

This paper presents HoliTom a training-free token merging framework for video LLMs. It performs an integration of outer-LLM spatio-temporal reduction, reducing initial token counts, and an inner-LLM token merging mechanism.  The paper received 1 borderline reject, 2 borderline accept, and 1 accept final ratings.  The reviewers are positive about the novelty of the approach and good performance.  Concerns regarding novelty compared to training-based methods and missing experiments were mostly addressed adequately by the rebuttal.  The borderline reject reviewer was not satisfied with the rebuttal, but was vague in their reason ("Some explanations are still unclear to me. Many details are missing in the paper and rebuttal.").  The AC deemed the rebuttal to address their main concerns, especially with the experiments using Qwen2.5-VL-7B.  After carefully considering the paper, reviews, and discussion, the AC recommends accept.